# First-Order Algorithms for Min-Max Optimization in Geodesic Metric Spaces

**Michael I. Jordan**     **Tianyi Lin**     **Emmanouil V. Vlatakis-Gkaragkounis**

University of California, Berkeley

{jordan@cs,darren_lin@,emvlatakis@}.berkeley.edu

## Abstract

From optimal transport to robust dimensionality reduction, a plethora of machine learning applications can be cast into the min-max optimization problems over Riemannian manifolds. Though many min-max algorithms have been analyzed in the Euclidean setting, it has proved elusive to translate these results to the Riemannian case. Zhang et al. have recently shown that geodesic convex concave Riemannian problems always admit saddle-point solutions. Inspired by this result, we study whether a performance gap between Riemannian and optimal Euclidean space convex-concave algorithms is necessary. We answer this question in the negative—we prove that the Riemannian corrected extragradient (RCEG) method achieves last-iterate convergence at a linear rate in the geodesically strongly-convex-concave case, matching the Euclidean result. Our results also extend to the stochastic or non-smooth case where RCEG and Riemanian gradient ascent descent (RGDA) achieve near-optimal convergence rates up to factors depending on curvature of the manifold.

## 1   Introduction

Constrained optimization problems arise throughout machine learning, in classical settings such as dimension reduction [2], dictionary learning [3, 4], and deep neural networks [5], but also in emerging problems involving decision-making and multi-agent interactions. While simple convex constraints (such as norm constraints) can be easily incorporated in standard optimization formulations, notably (proximal) gradient descent [6–10], in a range of other applications such as matrix recovery [11, 12], low-rank matrix factorization [13] and generative adversarial nets [14], the constraints are fundamentally nonconvex and are often treated via special heuristics.

Thus, a general goal is to design algorithms that systematically take account of special geometric structure of the feasible set [15–17]. A long line of work in the machine learning (ML) community has focused on understanding the geometric properties of commonly used constraints and how they affect optimization; [see, e.g., 18–26]. A prominent aspect of this agenda has been the re-expression of these constraints through the lens of Riemannian manifolds. This has given rise to new algorithms [27, 28] with a wide range of ML applications, inclduing online principal component analysis (PCA), the computation of Mahalanobis distance from noisy measurements [29], consensus distributed algorithms for aggregation in ad-hoc wireless networks [30] and maximum likelihood estimation for certain non-Gaussian (heavy- or light-tailed) distributions [31].

Going beyond simple minimization problems, the robustification of many ML tasks can be formulated as min-max optimization problems. Well-known examples in this domain include adversarial machine learning [32, 33], optimal transport [34], and online learning [9, 35, 36]. Similar to their minimization counterparts, non-convex constraints have been widely applicable to the min-max optimization as well [37–41]. Recently there has been significant effort in proving tighter results either under more structured assumptions [42–52], and/or obtaining last-iterate convergence guarantees [38, 40, 48, 53–

62] for computing min-max solutions in convex-concave settings. Nonetheless, the analysis of the iteration complexity in the general *non-convex non-concave* setting is still in its infancy [63, 64]. In response, the optimization community has recently studied how to extend standard min-max optimization algorithms such as gradient descent ascent (GDA) and extragradient (EG) to the Riemannian setting. In mathematical terms, given two Riemannian manifolds $\mathcal{M}, \mathcal{N}$ and a function $f : \mathcal{M} \times \mathcal{N} \to \mathbb{R}$, the Riemannian min-max optimization (RMMO) problem becomes

$$\min_{x \in \mathcal{M}} \max_{y \in \mathcal{N}} f(x, y).$$

The change of geometry from Euclidean to Riemannian poses several difficulties. Indeed, a fundamental stumbling block has been that this problem may not even have theoretically meaningful solutions. In contrast with minimization where an optimal solution in a bounded domain is always guaranteed [65], existence of such saddle points necessitates typically the application of topological fixed point theorems [66, 67], KKM Theory [68]). For the case of convex-concave $f$ with compact sets $\mathcal{X}$ and $\mathcal{Y}$, Sion [69] generalized the celebrated theorem [70] and guaranteed that a solution $(x^\star, y^\star)$ with the following property exists

$$\min_{x \in \mathcal{X}} f(x, y^\star) = f(x^\star, y^\star) = \max_{y \in \mathcal{Y}} f(x^\star, y).$$

However, at the core of the proof of this result is an ingenuous application of Helly's lemma [71] for the sublevel sets of $f$, and, until the work of Ivanov [72], it has been unclear how to formulate an analogous lemma for the Riemannian geometry. As a result, until recently have extensions of the min-max theorem been established, and only for restricted manifold families [73–75].

Zhang et al. [1] was the first to establish a min-max theorem for a flurry of Riemannian manifolds equipped with unique geodesics. Notice that this family is not a mathematical artifact since it encompasses many practical applications of RMMO, including Hadamard and Stiefel ones used in PCA [76]. Intuitively, the unique geodesic between two points of a manifold is the analogue of the a linear segment between two points in convex set: For any two points $x_1, x_2 \in \mathcal{X}$, their connecting geodesic is the unique shortest path contained in $\mathcal{X}$ that connects them.

Even when the RMMO is well defined, transferring the guarantees of traditional min-max optimization algorithms like Gradient Ascent Descent (GDA) and Extra-Gradient (EG) to the Riemannian case is non-trivial. Intuitively speaking, in the Euclidean realm the main leitmotif of the last-iterate analyses the aforementioned algorithms is a proof that $\delta_t = \|x_t - x^*\|^2$ is decreasing over time. To achieve this, typically the proof correlates $\delta_t$ and $\delta_{t-1}$ via a "square expansion," namely:

$$\underbrace{\|x_{t-1} - x^*\|^2}_{\alpha^2} = \underbrace{\|x_t - x^*\|^2}_{\beta^2} + \underbrace{\|x_{t-1} - x_t\|^2}_{\gamma^2} - \underbrace{2\langle x_t - x^*, x_{t-1} - x_t \rangle}_{2\beta\gamma \cos(\hat{A})}. \tag{1}$$

Notice, however that the above expression relies strongly on properties of Euclidean geometry (and the flatness of the corresponding line), namely that the the lines connecting the three points $x_t$, $x_{t-1}$ and $x^*$ form a triangle; indeed, it is the generalization of the Pythagorean theorem, known also as the law of cosines, for the induced triangle $(ABC) := \{(x_t, x_{t-1}, x^*)\}$. In a uniquely geodesic manifold such triangle may not belong to the manifold as discussed above. As a result, the difference of distances to the equilibrium using the geodesic paths $d^2_{\mathcal{M}}(x_t, x^*) - d^2_{\mathcal{M}}(x_{t-1}, x^*)$ generally cannot be given in a closed form. The manifold's curvature controls how close these paths are to forming a Euclidean triangle. In fact, the phenomenon of *distance distortion*, as it is typically called, was hypothesised by Zhang et al. [1, Section 4.2] to be the cause of exponential slowdowns when applying EG to RMMO problems when compared to their Euclidean counterparts.

Multiple attempts have been made to bypass this hurdle. Huang et al. [77] analyzed the Riemannian GDA (RGDA) for the non-convex non-concave setting. However, they do not present any last-iterate convergence results and, even in the average/best iterate setting, they only derive sub-optimal rates for the geodesic convex-concave setting due to the lack of the machinery that convex analysis and optimization offers they derive sub-optimal rates for the geodesic convex-concave case, which is the problem of our interest. The analysis of Han et al. [78] for Riemannian Hamiltonian Method (RHM), matches the rate of second-order methods in the Euclidean case. Although theoretically faster in terms of iterations, second-order methods are not preferred in practice since evaluating second order derivatives for optimization problems of thousands to millions of parameters quickly becomes prohibitive. Finally, Zhang et al. [1] leveraged the standard averaging output trick in EG to derive a sublinear convergence rate of $O(1/\epsilon)$ for the general geodesically convex-concave Riemannian

framework. In addition, they conjectured that the use of a different method could close the exponential gap for the geodesically strongly-convex-strongly-convex scenario and its Euclidean counterpart.

Given this background, a crucial question underlying the potential for successful application of first-order algorithms to Riemannian settings is the following:

*Is a performance gap necessary between Riemannian and Euclidean optimal convex-concave algorithms in terms of accuracy and the condition number?*

## 1.1 Our Contributions

Our aim in this paper is to provide an extensive analysis of the Riemannian counterparts of Euclidean optimal first-order methods adapted to the manifold-constrained setting. For the case of the smooth objectives, we consider the *Riemannian corrected extragradient* (RCEG) method while for non-smooth cases, we analyze the textbook *Riemannian gradient descent ascent* (RGDA) method. Our main results are summarized in the following table.

| Perf. Measure | Setting | Complexity | Theorem |
|:---:|:---:|:---:|:---:|
| \multicolumn | | Alg: *RCEG*. Smooth setting with $\ell$-*Lipschitz Gradient* (cf. Assumption 2.1, 3.1 and 3.2) | |
| Last-Iterate | Det. GSCSC | $O\left(\kappa(\sqrt{\tau_0} + \frac{1}{\underline{\xi}_0})\log(\frac{1}{\epsilon})\right)$ | **Thm.** 3.1 |
| Last-Iterate | Stoc. GSCSC | $O\left(\kappa(\sqrt{\tau_0} + \frac{1}{\underline{\xi}_0})\log(\frac{1}{\epsilon}) + \frac{\sigma^2 \overline{\xi}_0}{\mu^2 \epsilon}\log(\frac{1}{\epsilon})\right)$ | **Thm.** 3.2 |
| Avg-Iterate | Det. GCC | $O\left(\frac{\ell\sqrt{\tau_0}}{\epsilon}\right)$ | [1, **Thm.1**] |
| Avg-Iterate | Stoc. GCC | $O\left(\frac{\ell\sqrt{\tau_0}}{\epsilon} + \frac{\sigma^2 \overline{\xi}_0}{\epsilon^2}\right)$ | **Thm.** 3.3 |
| | | Alg: *RGDA*. Nonsmooth setting with $L$-*Lipschitz Function* (cf. Assumption D.1 and D.2) | |
| Last-Iterate | Det. GSCSC | $O\left(\frac{L^2 \overline{\xi}_0}{\mu^2 \epsilon}\right)$ | **Thm.** D.1 |
| Last-Iterate | Stoc. GSCSC | $O\left(\frac{(L^2+\sigma^2)\overline{\xi}_0}{\mu^2 \epsilon}\right)$ | **Thm.** D.3 |
| Avg-Iterate | Det. GCC | $O\left(\frac{L^2 \overline{\xi}_0}{\epsilon^2}\right)$ | **Thm.** D.2 |
| Avg-Iterate | Stoc. GCC | $O\left(\frac{(L^2+\sigma^2)\overline{\xi}_0}{\epsilon^2}\right)$ | **Thm.** D.4 |

For the definition of the acronyms, Det and Stoc stand for deterministic and stochastic, respectively. GSCSC and GCC stand for geodesically strongly-convex-strongly-concave (cf. Assumption 3.1 or Assumption D.1) and geodesically convex-concave (cf. Assumption 3.2 or Assumption D.2). Here $\epsilon \in (0,1)$ is the accuracy, $L, \ell$ the Lipschitzness of the objective and its gradient, $\kappa = \ell/\mu$ is the condition number of the function, where $\mu$ is the strong convexity parameter, $(\tau_0, \underline{\xi}_0, \overline{\xi}_0)$ are curvature parameters (cf. Assumption 2.1), and $\sigma^2$ is the variance of a Riemannian gradient estimator.

Our first main contribution is the derivation of a linear convergence rate for RCEG, answering the open conjecture of [1] about the performance gap of single-loop extragradient methods. Indeed, while a direct comparison between $d^2_{\mathcal{M}}(x_t, x^*)$ and $d^2_{\mathcal{M}}(x_{t-1}, x^*)$ is infeasible, we are able to establish a relationship between the iterates via appeal to the duality gap function and obtain a contraction in terms of $d^2_{\mathcal{M}}(x_t, x^*)$. In other words, the effect of Riemannian distance distortion is quantitative (the contraction ratio will depend on it) rather than qualitative (the geometric contraction still remains under a proper choice of constant stepsize). More specifically, we use $d^2_{\mathcal{M}}(x_t, x^*) + d^2_{\mathcal{N}}(y_t, y^*)$ and $d^2_{\mathcal{M}}(x_{t+1}, x^*) + d^2_{\mathcal{N}}(y_{t+1}, y^*)$ to bound a gap function defined by $f(\hat{x}_t, y^*) - f(x^*, \hat{y}_t)$. Since the objective function is geodesically strongly-convex-strongly-concave, we have $f(\hat{x}_t, y^*) - f(x^*, \hat{y}_t)$ is lower bounded by $\frac{\mu}{2}(d_{\mathcal{M}}(\hat{x}_t, x^*)^2 + d_{\mathcal{N}}(\hat{y}_t, y^*)^2)$. Then, using the relationship between $(x_t, y_t)$ and $(\hat{x}_t, \hat{y}_t)$, we conclude the desired results in Theorem 3.1. Notably, our approach is not affected by the nonlinear geometry of the manifold.

Secondly, we endeavor to give a systematic analysis of aspects of the objective function, including its smoothness, its convexity and oracle access. As we shall see, similar to the Euclidean case, better finite-time convergence guarantees are connected with a geodesic smoothness condition. For the sake of completeness, in the paper's supplement we present the performance of Riemannian GDA for the full spectrum of stochasticity for the non-smooth case. More specifically, for the stochastic setting, the key ingredient to get the optimal convergence rate is to carefully select the step size such that the noise of the gradient estimator will not affect the final convergence rate significantly. As a highlight, such

technique has been used for analyzing stochastic RCEG in the Euclidean setting [79] and our analysis can be seen as the extension to the Riemannian setting. For the nonsmooth setting, the analysis is relatively simpler compared to smooth settings but we still need to deal with the issue caused by the nonlinear geometry of manifolds and the interplay between the distortion of Riemannian metrics, the gap function and the bounds of Lipschitzness of our bi-objective. Interestingly, the rates we derive are near optimal in terms of accuracy and condition number of the objective, and analogous to their Euclidean counterparts.

## 2 Preliminaries and Technical Background

We present the basic setup and optimality conditions for Riemannian min-max optimization. Indeed, we focus on some of key concepts that we need from Riemannian geometry, deferring a fuller presentation, including motivating examples and further discussion of related work, to Appendix A-C.

**Riemannian geometry.** An $n$-dimensional manifold $\mathcal{M}$ is a topological space where any point has a neighborhood that is homeomorphic to the $n$-dimensional Euclidean space. For each $x \in \mathcal{M}$, each tangent vector is tangent to all parametrized curves passing through $x$ and the tangent space $T_x\mathcal{M}$ of a manifold $\mathcal{M}$ at this point is defined as the set of all tangent vectors. A Riemannian manifold $\mathcal{M}$ is a smooth manifold that is endowed with a smooth ("Riemannian") metric $\langle \cdot, \cdot \rangle_x$ on the tangent space $T_x\mathcal{M}$ for each point $x \in \mathcal{M}$. The inner metric induces a norm $\| \cdot \|_x$ on the tangent spaces.

A geodesic can be seen as the generalization of an Euclidean linear segment and is modeled as a smooth curve (map), $\gamma : [0, 1] \mapsto \mathcal{M}$, which is locally a distance minimizer. Additionally, because of the non-flatness of a manifold a different relation between the angles and the lengths of an arbitrary geodesic triangle is induced. This distortion can be quantified via the *sectional curvature* parameter $\kappa_\mathcal{M}$ thanks to Toponogov's theorem [80, 81]. A constructive consequence of this definition are the

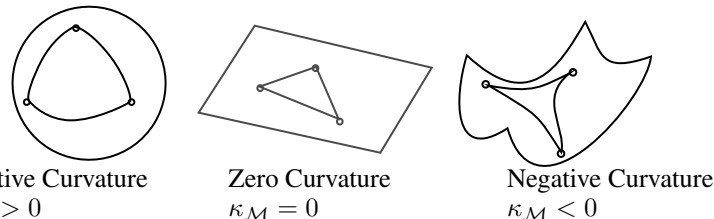

| Positive Curvature | Zero Curvature | Negative Curvature |
| $\kappa_\mathcal{M} > 0$ | $\kappa_\mathcal{M} = 0$ | $\kappa_\mathcal{M} < 0$ |

trigonometric comparison inequalities (TCIs) that will be essential in our proofs; see Alimisis et al. [82, Corollary 2.1] and Zhang and Sra [83, Lemma 5] for detailed derivations. Assuming bounded sectional curvature, TCIs provide a tool for bounding Riemannian "inner products" that are more troublesome than classical Euclidean inner products.

The following proposition summarizes the TCIs that we will need; note that if $\kappa_\mathrm{min} = \kappa_\mathrm{max} = 0$ (i.e., Euclidean spaces), then the proposition reduces to the law of cosines.

**Proposition 2.1** *Suppose that $\mathcal{M}$ is a Riemannian manifold and let $\Delta$ be a geodesic triangle in $\mathcal{M}$ with the side length $a$, $b$, $c$ and let $A$ be the angle between $b$ and $c$. Then, we have*

*1. If $\kappa_\mathcal{M}$ that is upper bounded by $\kappa_\mathrm{max} > 0$ and the diameter of $\mathcal{M}$ is bounded by $\frac{\pi}{\sqrt{\kappa_\mathrm{max}}}$, then*

$$a^2 \geq \underline{\xi}(\kappa_\mathrm{max}, c) \cdot b^2 + c^2 - 2bc\cos(A),$$

*where $\underline{\xi}(\kappa, c) := 1$ for $\kappa \leq 0$ and $\underline{\xi}(\kappa, c) := c\sqrt{\kappa}\cot(c\sqrt{\kappa}) < 1$ for $\kappa > 0$.*
*2. If $\kappa_\mathcal{M}$ is lower bounded by $\kappa_\mathrm{min}$, then*

$$a^2 \leq \overline{\xi}(\kappa_\mathrm{min}, c) \cdot b^2 + c^2 - 2bc\cos(A),$$

*where $\overline{\xi}(\kappa, c) := c\sqrt{-\kappa}\coth(c\sqrt{-\kappa}) > 1$ if $\kappa < 0$ and $\overline{\xi}(\kappa, c) := 1$ if $\kappa \geq 0$.*

Also, in contrast to the Euclidean case, $x$ and $v = \mathrm{grad}_x f(x)$ do not lie in the same space, since $\mathcal{M}$ and $T_x\mathcal{M}$ respectively are distinct entities. The interplay between these dual spaces typically is carried out via the *exponential maps*. An exponential map at a point $x \in \mathcal{M}$ is a mapping from the tangent space $T_x\mathcal{M}$ to $\mathcal{M}$. In particular, $y := \mathrm{Exp}_x(v) \in \mathcal{M}$ is defined such that there exists a geodesic $\gamma : [0, 1] \mapsto \mathcal{M}$ satisfying $\gamma(0) = x$, $\gamma(1) = y$ and $\gamma'(0) = v$. The inverse

map exists since the manifold has a unique geodesic between any two points, which we denote as $\mathrm{Exp}_x^{-1} : \mathcal{M} \mapsto T_x\mathcal{M}$. Accordingly, we have $d_{\mathcal{M}}(x, y) = \|\mathrm{Exp}_x^{-1}(y)\|_x$ is the Riemannian distance induced by the exponential map.

Finally, in contrast again to Euclidean spaces, we cannot compare the tangent vectors at different points $x, y \in \mathcal{M}$ since these vectors lie in different tangent spaces. To resolve this issue, it suffices to define a transport mapping that moves a tangent vector along the geodesics and also preserves the length and Riemannian metric $\langle \cdot, \cdot \rangle_x$; indeed, we can define a parallel transport $\Gamma_x^y : T_x\mathcal{M} \mapsto T_y\mathcal{M}$ such that the inner product between any $u, v \in T_x\mathcal{M}$ is preserved; i.e., $\langle u, v \rangle_x = \langle \Gamma_x^y(u), \Gamma_x^y(v) \rangle_y$.

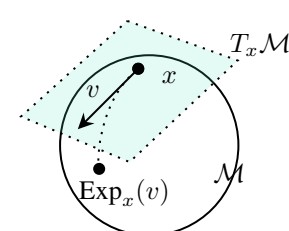

**Riemannian min-max optimization and function classes.** We let $\mathcal{M}$ and $\mathcal{N}$ be Riemannian manifolds with unique geodesic and bounded sectional curvature and assume that the function $f : \mathcal{M} \times \mathcal{N} \mapsto \mathbb{R}$ is defined on the product of these manifolds. The regularity conditions that we impose on the function $f$ are as follows.

**Definition 2.1** *A function $f : \mathcal{M} \times \mathcal{N} \mapsto \mathbb{R}$ is* geodesically $L$-Lipschitz *if for $\forall x, x' \in \mathcal{M}$ and $\forall y, y' \in \mathcal{N}$, the following statement holds true: $|f(x, y) - f(x', y')| \leq L(d_{\mathcal{M}}(x, x') + d_{\mathcal{N}}(y, y'))$. Additionally, if function $f$ is also differentiable, it is called* geodesically $\ell$-smooth *if for $\forall x, x' \in \mathcal{M}$ and $\forall y, y' \in \mathcal{N}$, the following statement holds true,*

$$
\begin{aligned}
\|\mathrm{grad}_x f(x, y) - \Gamma_{x'}^x \mathrm{grad}_x f(x', y')\| &\leq \ell(d_{\mathcal{M}}(x, x') + d_{\mathcal{N}}(y, y')), \\
\|\mathrm{grad}_y f(x, y) - \Gamma_{y'}^y \mathrm{grad}_y f(x', y')\| &\leq \ell(d_{\mathcal{M}}(x, x') + d_{\mathcal{N}}(y, y')),
\end{aligned}
$$

*where $(\mathrm{grad}_x f(x', y'), \mathrm{grad}_y f(x', y')) \in T_{x'}\mathcal{M} \times T_{y'}\mathcal{N}$ is the Riemannian gradient of $f$ at $(x', y')$, $\Gamma_{x'}^x$ is the parallel transport of $\mathcal{M}$ from $x'$ to $x$, and $\Gamma_{y'}^y$ is the parallel transport of $\mathcal{N}$ from $y'$ to $y$.*

**Definition 2.2** *A function $f : \mathcal{M} \times \mathcal{N} \to \mathbb{R}$ is* geodesically strongly-convex-strongly-concave *with the modulus $\mu > 0$ if the following statement holds true,*

$$
\begin{aligned}
f(x', y) &\geq f(x, y) + \langle \mathrm{subgrad}_x f(x, y), \mathrm{Exp}_x^{-1}(x') \rangle_x + \tfrac{\mu}{2}(d_{\mathcal{M}}(x, x'))^2, &&\text{for each } y \in \mathcal{N}, \\
f(x, y') &\leq f(x, y) + \langle \mathrm{subgrad}_y f(x, y), \mathrm{Exp}_y^{-1}(y') \rangle_y - \tfrac{\mu}{2}(d_{\mathcal{N}}(y, y'))^2, &&\text{for each } x \in \mathcal{M}.
\end{aligned}
$$

*where $(\mathrm{subgrad}_x f(x', y'), \mathrm{subgrad}_y f(x', y')) \in T_{x'}\mathcal{M} \times T_{y'}\mathcal{N}$ is a Riemannian subgradient of $f$ at a point $(x', y')$. A function $f$ is* geodesically convex-concave *if the above holds true with $\mu = 0$.*

Following standard conventions in Riemannian optimization [1, 82, 83], we make the following assumptions on the manifolds and objective functions:[1]

**Assumption 2.1** *The objective function $f : \mathcal{M} \times \mathcal{N} \mapsto \mathbb{R}$ and manifolds $\mathcal{M}$ and $\mathcal{N}$ satisfy*

1. *The diameter of the domain $\{(x, y) \in \mathcal{M} \times \mathcal{N} : -\infty < f(x, y) < +\infty\}$ is bounded by $D > 0$.*
2. *$\mathcal{M}, \mathcal{N}$ admit unique geodesic paths for any $(x, y), (x', y') \in \mathcal{M} \times \mathcal{N}$.*
3. *The sectional curvatures of $\mathcal{M}$ and $\mathcal{N}$ are both bounded in the range $[\kappa_{\min}, \kappa_{\max}]$ with $\kappa_{\min} \leq 0$. If $\kappa_{\max} > 0$, we assume that the diameter of manifolds is bounded by $\frac{\pi}{\sqrt{\kappa_{\max}}}$.*

Under these conditions, Zhang et al. [1] proved an analog of Sion's minimax theorem [69] in geodesic metric spaces. Formally, we have

$$
\max_{y \in \mathcal{N}} \min_{x \in \mathcal{M}} f(x, y) = \min_{x \in \mathcal{M}} \max_{y \in \mathcal{N}} f(x, y),
$$

which guarantees that there exists at least one global saddle point $(x^\star, y^\star) \in \mathcal{M} \times \mathcal{N}$ such that $\min_{x \in \mathcal{M}} f(x, y^\star) = f(x^\star, y^\star) = \max_{y \in \mathcal{Y}} f(x^\star, y)$. Note that the unicity of geodesics assumption is algorithm-independent and is imposed for guaranteeing that a saddle-point solution always exist. Even though this rules out many manifolds of interest, there are still many manifolds that satisfy such conditions. More specifically, the Hadamard manifold (manifolds with non-positive curvature, $\kappa_{\max} = 0$) has a unique geodesic between any two points. This also becomes a common regularity condition in Riemannian optimization [82, 83]. For any point $(\hat{x}, \hat{y}) \in \mathcal{M} \times \mathcal{N}$, the duality gap $f(\hat{x}, y^\star) - f(x^\star, \hat{y})$ thus gives an optimality criterion.

---

[1]In particular, our assumed upper and lower bounds $\kappa_{\min}, \kappa_{\max}$ guarantee that TCIs in Proposition 2.1 can be used in our analysis for proving finite-time convergence.

| **Algorithm 1** RCEG | **Algorithm 2** SRCEG |
|---|---|
| **Input:** initial points $(x_0, y_0)$ and stepsizes $\eta > 0$. | **Input:** initial points $(x_0, y_0)$ and stepsizes $\eta > 0$. |
| **for** $t = 0, 1, 2, \ldots, T-1$ **do** | **for** $t = 0, 1, 2, \ldots, T-1$ **do** |
|     Query $(g_x^t, g_y^t) \leftarrow (\text{grad}_x f(x_t, y_t), \text{grad}_y f(x_t, y_t))$, |     Query $(g_x^t, g_y^t)$ as a **noisy** estimator of Riemannian gradient of $f$ at a point $(x_t, y_t)$. |
| the Riemannian gradient of $f$ at a point $(x_t, y_t)$ | |
|     $\hat{x}_t \leftarrow \text{Exp}_{x_t}(-\eta \cdot g_x^t)$. |     $\hat{x}_t \leftarrow \text{Exp}_{x_t}(-\eta \cdot g_x^t)$. |
|     $\hat{y}_t \leftarrow \text{Exp}_{y_t}(\eta \cdot g_y^t)$. |     $\hat{y}_t \leftarrow \text{Exp}_{y_t}(\eta \cdot g_y^t)$. |
|     Query $(\hat{g}_x^t, \hat{g}_y^t) \leftarrow (\text{grad}_x f(\hat{x}_t, \hat{y}_t), \text{grad}_y f(\hat{x}_t, \hat{y}_t))$, |     Query $(\hat{g}_x^t, \hat{g}_y^t)$ as a **noisy** estimator of Riemannian gradient of $f$ at a point $(\hat{x}_t, \hat{y}_t)$. |
| the Riemannian gradient of $f$ at a point $(\hat{x}_t, \hat{y}_t)$ | |
|     $x_{t+1} \leftarrow \text{Exp}_{\hat{x}_t}(-\eta \cdot \hat{g}_x^t + \text{Exp}_{\hat{x}_t}^{-1}(x_t))$. |     $x_{t+1} \leftarrow \text{Exp}_{\hat{x}_t}(-\eta \cdot \hat{g}_x^t + \text{Exp}_{\hat{x}_t}^{-1}(x_t))$. |
|     $y_{t+1} \leftarrow \text{Exp}_{\hat{y}_t}(\eta \cdot \hat{g}_y^t + \text{Exp}_{\hat{y}_t}^{-1}(y_t))$. |     $y_{t+1} \leftarrow \text{Exp}_{\hat{y}_t}(\eta \cdot \hat{g}_y^t + \text{Exp}_{\hat{y}_t}^{-1}(y_t))$. |
| **end for** | **end for** |

**Definition 2.3** *A point* $(\hat{x}, \hat{y}) \in \mathcal{M} \times \mathcal{N}$ *is an* $\epsilon$-saddle point *of a geodesically convex-concave function* $f(\cdot, \cdot)$ *if* $f(\hat{x}, y^\star) - f(x^\star, \hat{y}) \leq \epsilon$ *where* $(x^\star, y^\star) \in \mathcal{M} \times \mathcal{N}$ *is a global saddle point.*

In the setting where $f$ is geodesically strongly-convex-strongly-concave with $\mu > 0$, it is not difficult to verify the uniqueness of a global saddle point $(x^\star, y^\star) \in \mathcal{M} \times \mathcal{N}$. Then, we can consider the distance gap $(d(\hat{x}, x^\star))^2 + (d(\hat{y}, y^\star))^2$ as an optimality criterion for any point $(\hat{\mathbf{x}}, \hat{\mathbf{y}}) \in \mathcal{M} \times \mathcal{N}$.

**Definition 2.4** *A point* $(\hat{x}, \hat{y}) \in \mathcal{M} \times \mathcal{N}$ *is an* $\epsilon$-saddle point of a geodesically strongly-convex-strongly-concave function $f(\cdot, \cdot)$ if $(d(\hat{x}, x^\star))^2 + (d(\hat{y}, y^\star))^2 \leq \epsilon$, where $(x^\star, y^\star) \in \mathcal{M} \times \mathcal{N}$ is a global saddle point. If $f$ is also geodesically $\ell$-smooth, we denote $\kappa = \frac{\ell}{\mu}$ as the condition number.

Given the above definitions, we can ask whether it is possible to find an $\epsilon$-saddle point efficiently or not. In this context, Zhang et al. [1] have answered this question in the affirmative for the setting where $f$ is geodesically $\ell$-smooth and geodesically convex-concave; indeed, they derive the convergence rate of Riemannian corrected extragradient (RCEG) method in terms of time-average iterates and also conjecture that *RCEG does not guarantee convergence at a linear rate in terms of last iterates when $f$ is geodesically $\ell$-smooth and geodesically strongly-convex-strongly-concave, due to the existence of distance distortion*; see Zhang et al. [1, Section 4.2]. Surprisingly, we show in Section 3 that RCEG with constant stepsize can achieve last-iterate convergence at a linear rate. Moreover, we establish the optimal convergence rates of stochastic RCEG for certain choices of stepsize for both geodesically convex-concave and geodesically strongly-convex-strongly-concave settings.

## 3 Riemannian Corrected Extragradient Method

In this section, we revisit the scheme of Riemannian corrected extragradient (RCEG) method proposed by Zhang et al. [1] and extend it to a stochastic algorithm that we refer to as *stochastic RCEG*. We present our main results on an optimal last-iterate convergence guarantee for the geodesically strongly-convex-strongly-concave setting (both deterministic and stochastic) and a time-average convergence guarantee for the geodesically convex-concave setting (stochastic). This complements the time-average convergence guarantee for geodesically convex-concave setting (deterministic) [1, Theorem 4.1] and resolves an open problem posted in Zhang et al. [1, Section 4.2].

### 3.1 Algorithmic scheme

The recently proposed *Riemannian corrected extragradient* (RCEG) method [1] is a natural extension of the celebrated extragradient (EG) method to the Riemannian setting. Its scheme resembles that of EG in Euclidean spaces but employs a simple modification in the extrapolation step to accommodate the nonlinear geometry of Riemannian manifolds. Let us provide some intuition how such modifications work.

We start with a basic version of EG as follows, where $\mathcal{M}$ and $\mathcal{N}$ are classically restricted to be convex constraint sets in Euclidean spaces:

$$\begin{aligned} \hat{x}_t &\leftarrow \text{proj}_{\mathcal{M}}(x_t - \eta \cdot \nabla_x f(x_t, y_t)), & \hat{y}_t &\leftarrow \text{proj}_{\mathcal{N}}(y_t + \eta \cdot \nabla_y f(x_t, y_t)), \\ x_{t+1} &\leftarrow \text{proj}_{\mathcal{M}}(x_t - \eta \cdot \nabla_x f(\hat{x}_t, \hat{y}_t)), & y_{t+1} &\leftarrow \text{proj}_{\mathcal{N}}(y_t + \eta \cdot \nabla_y f(\hat{x}_t, \hat{y}_t)). \end{aligned} \quad (2)$$

Turning to the setting where $\mathcal{M}$ and $\mathcal{N}$ are Riemannian manifolds, the rather straightforward way to do the generalization is to replace the projection operator by the corresponding exponential map and the gradient by the corresponding Riemannian gradient. For the first line of Eq. (2), this approach works and leads to the following updates:

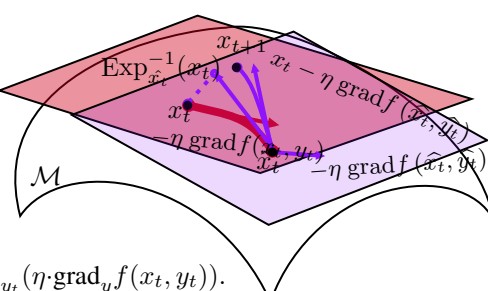

$$\hat{x}_t \leftarrow \mathrm{Exp}_{x_t}(-\eta \cdot \mathrm{grad}_x f(x_t, y_t)), \quad \hat{y}_t \leftarrow \mathrm{Exp}_{y_t}(\eta \cdot \mathrm{grad}_y f(x_t, y_t)).$$

However, we encounter some issues for the second line of Eq. (2): The aforementioned approach leads to some problematic updates, $x_{t+1} \leftarrow \mathrm{Exp}_{x_t}(-\eta \cdot \mathrm{grad}_x f(\hat{x}_t, \hat{y}_t))$ and $y_{t+1} \leftarrow \mathrm{Exp}_{y_t}(\eta \cdot \mathrm{grad}_y f(\hat{x}_t, \hat{y}_t))$; indeed, the exponential maps $\mathrm{Exp}_{x_t}(\cdot)$ and $\mathrm{Exp}_{y_t}(\cdot)$ are defined from $T_{x_t}\mathcal{M}$ to $\mathcal{M}$ and from $T_{y_t}\mathcal{N}$ to $\mathcal{N}$ respectively. However, we have $-\mathrm{grad}_x f(\hat{x}_t, \hat{y}_t) \in T_{\hat{x}_t}\mathcal{M}$ and $\mathrm{grad}_y f(\hat{x}_t, \hat{y}_t) \in T_{\hat{y}_t}\mathcal{N}$. This motivates us to reformulate the second line of Eq. (2) as follows:

$$x_{t+1} \leftarrow \mathrm{proj}_{\mathcal{M}}(\hat{x}_t - \eta \cdot \nabla_x f(\hat{x}_t, \hat{y}_t) + (x_t - \hat{x}_t)), \quad y_{t+1} \leftarrow \mathrm{proj}_{\mathcal{N}}(\hat{y}_t + \eta \cdot \nabla_y f(\hat{x}_t, \hat{y}_t) + (y_t - \hat{y}_t)).$$

In the general setting of Riemannian manifolds, the terms $x_t - \hat{x}_t$ and $y_t - \hat{y}_t$ become $\mathrm{Exp}_{\hat{x}_t}^{-1}(x_t) \in T_{\hat{x}_t}\mathcal{M}$ and $\mathrm{Exp}_{\hat{y}_t}^{-1}(y_t) \in T_{\hat{y}_t}\mathcal{N}$. This observation yields the following updates:

$$x_{t+1} \leftarrow \mathrm{Exp}_{\hat{x}_t}(-\eta \cdot \mathrm{grad}_x f(\hat{x}_t, \hat{y}_t) + \mathrm{Exp}_{\hat{x}_t}^{-1}(x_t)), \quad \hat{y}_t \leftarrow \mathrm{Exp}_{\hat{y}_t}(\eta \cdot \mathrm{grad}_y f(\hat{x}_t, \hat{y}_t) + \mathrm{Exp}_{\hat{y}_t}^{-1}(y_t)).$$

We summarize the resulting RCEG method in Algorithm 1 and present the stochastic extension with noisy estimators of Riemannian gradients of $f$ in Algorithm 2.

## 3.2 Main results

We present our main results on global convergence for Algorithms 1 and 2. To simplify the presentation, we treat separately the following two cases:

**Assumption 3.1** *The objective function $f$ is geodesically $\ell$-smooth and geodesically strongly-convex-strongly-concave with $\mu > 0$.*

**Assumption 3.2** *The objective function $f$ is geodesically $\ell$-smooth and geodesically convex-concave.*

Letting $(x^\star, y^\star) \in \mathcal{M} \times \mathcal{N}$ be a global saddle point of $f$ (which exists under either Assumption 3.1 or 3.2), we let $D_0 = (d_{\mathcal{M}}(x_0, x^\star))^2 + (d_{\mathcal{N}}(y_0, y^\star))^2 > 0$ and $\kappa = \ell/\mu$ for geodesically strongly-convex-strongly-concave setting. For simplicity of presentation, we also define a ratio $\tau(\cdot, \cdot)$ that measures how non-flatness changes in the spaces: $\tau([\kappa_{\min}, \kappa_{\max}], c) = \frac{\overline{\xi}(\kappa_{\min}, c)}{\underline{\xi}(\kappa_{\max}, c)} \geq 1$. We summarize our results for Algorithm 1 in the following theorem.

**Theorem 3.1** *Given Assumptions 2.1 and 3.1, and letting $\eta = \min\{1/(2\ell\sqrt{\tau_0}), \underline{\xi}_0/(2\mu)\}$, there exists some $T > 0$ such that the output of Algorithm 1 satisfies that $(d(x_T, x^\star))^2 + (d(y_T, y^\star))^2 \leq \epsilon$ (i.e., an $\epsilon$-saddle point of $f$ in Definition 2.4) and the total number of Riemannian gradient evaluations is bounded by*

$$O\left(\left(\kappa\sqrt{\tau_0} + \frac{1}{\underline{\xi}_0}\right)\log\left(\frac{D_0}{\epsilon}\right)\right),$$

*where $\tau_0 = \tau([\kappa_{\min}, \kappa_{\max}], D) \geq 1$ measures how non-flatness changes in $\mathcal{M}$ and $\mathcal{N}$ and $\underline{\xi}_0 = \underline{\xi}(\kappa_{\max}, D) \leq 1$ is properly defined in Proposition 2.1.*

**Remark 3.1** *Theorem 3.1 illustrates the last-iterate convergence of Algorithm 1 for solving geodesically strongly-convex-strongly-concave problems, thereby resolving an open problem delineated by Zhang et al. [1]. Further, the dependence on $\kappa$ and $1/\epsilon$ cannot be improved since it matches the lower bound established for min-max optimization problems in Euclidean spaces [84]. However, we believe that the dependence on $\tau_0$ and $\underline{\xi}_0$ is not tight, and it is of interest to either improve the rate or establish a lower bound for general Riemannian min-max optimization.*

**Remark 3.2** *The current theoretical analysis covers local geodesic strong-convex-strong-concave settings. The key ingredient is how to define the local region; indeed, if we say the set of $\{(x, y) : d_{\mathcal{M}}(x, x^\star) \leq \delta, d_{\mathcal{N}}(y_t, y^\star) \leq \delta\}$ is a local region where the function is geodesic strong-convex-strong-concave. Then, the set of $\{(x, y) : (d_{\mathcal{M}}(x, x^\star)^2 + d_{\mathcal{N}}(y_t, y^\star)^2) \leq \delta^2\}$ must be contained in the above local region and the objective function is also geodesic strong-convex-strong-concave. If $(x_0, y_0) \in \{(x, y) : (d_{\mathcal{M}}(x, x^\star)^2 + d_{\mathcal{N}}(y_t, y^\star)^2) \leq \delta^2\}$, our theoretical analysis guarantees the last-iterate linear convergence rate. Such argument and definition of local region were standard for min-max optimization in the Euclidean setting; see Liang and Stokes [55, Assumption 2.1]. For an important optimization problem that is globally geodesically strongly-convex-strongly-concave, we refer to Appendix B where Robust matrix Karcher mean problem is indeed the desired one.*

In the scheme of SRECG, we highlight that $(g_x^t, g_y^t)$ and $(\hat{g}_x^t, \hat{g}_y^t)$ are noisy estimators of Riemannian gradients of $f$ at $(x_t, y_t)$ and $(\hat{x}_t, \hat{y}_t)$. It is necessary to impose the conditions such that these estimators are unbiased and has bounded variance. By abuse of notation, we assume that

$$
\begin{aligned}
g_x^t &= \mathrm{grad}_x f(x_t, y_t) + \xi_x^t, & g_y^t &= \mathrm{grad}_y f(x_t, y_t) + \xi_y^t, \\
\hat{g}_x^t &= \mathrm{grad}_x f(\hat{x}_t, \hat{y}_t) + \hat{\xi}_x^t, & \hat{g}_y^t &= \mathrm{grad}_y f(\hat{x}_t, \hat{y}_t) + \hat{\xi}_y^t.
\end{aligned} \tag{3}
$$

where the noises $(\xi_x^t, \xi_y^t)$ and $(\hat{\xi}_x^t, \hat{\xi}_y^t)$ are independent and satisfy that

$$
\begin{aligned}
\mathbb{E}[\xi_x^t] &= 0, & \mathbb{E}[\xi_y^t] &= 0, & \mathbb{E}[\|\xi_x^t\|^2 + \|\xi_y^t\|^2] &\leq \sigma^2, \\
\mathbb{E}[\hat{\xi}_x^t] &= 0, & \mathbb{E}[\hat{\xi}_y^t] &= 0, & \mathbb{E}[\|\hat{\xi}_x^t\|^2 + \|\hat{\xi}_y^t\|^2] &\leq \sigma^2.
\end{aligned} \tag{4}
$$

We are ready to summarize our results for Algorithm 2 in the following theorems.

**Theorem 3.2** *Given Assumptions 2.1 and 3.1, letting Eq. (3) and Eq. (4) hold with $\sigma > 0$ and letting $\eta > 0$ satisfy $\eta = \min\{\frac{1}{24\ell\sqrt{\tau_0}}, \frac{\underline{\xi}_0}{2\mu}, \frac{2(\log(T) + \log(\mu^2 D_0 \sigma^{-2}))}{\mu T}\}$, there exists some $T > 0$ such that the output of Algorithm 2 satisfies that $\mathbb{E}[(d(x_T, x^\star))^2 + (d(y_T, y^\star))^2] \leq \epsilon$ and the total number of noisy Riemannian gradient evaluations is bounded by*

$$
O\left( \left( \kappa\sqrt{\tau_0} + \frac{1}{\underline{\xi}_0} \right) \log\left( \frac{D_0}{\epsilon} \right) + \frac{\sigma^2 \overline{\xi}_0}{\mu^2 \epsilon} \log\left( \frac{1}{\epsilon} \right) \right),
$$

*where $\tau_0 = \tau([\kappa_{\min}, \kappa_{\max}], D) \geq 1$ measures how non-flatness changes in $\mathcal{M}$ and $\mathcal{N}$ and $\underline{\xi}_0 = \underline{\xi}(\kappa_{\max}, D) \leq 1$ is properly defined in Proposition 2.1.*

**Theorem 3.3** *Given Assumptions 2.1 and 3.2 and assume that Eq. (3) and Eq. (4) hold with $\sigma > 0$ and let $\eta > 0$ satisfies that $\eta = \min\{\frac{1}{4\ell\sqrt{\tau_0}}, \frac{1}{\sigma}\sqrt{\frac{D_0}{\overline{\xi}_0 T}}\}$, there exists some $T > 0$ such that the output of Algorithm 2 satisfies that $\mathbb{E}[f(\bar{x}_T, y^\star) - f(x^\star, \bar{y}_T)] \leq \epsilon$ and the total number of noisy Riemannian gradient evaluations is bounded by*

$$
O\left( \frac{\ell D_0 \sqrt{\tau_0}}{\epsilon} + \frac{\sigma^2 \overline{\xi}_0}{\epsilon^2} \right),
$$

*where $\tau_0 = \tau([\kappa_{\min}, \kappa_{\max}], D)$ measures how non-flatness changes in $\mathcal{M}$ and $\mathcal{N}$ and $\overline{\xi}_0 = \overline{\xi}(\kappa_{\min}, D) \geq 1$ is properly defined in Proposition 2.1. The time-average iterates $(\bar{x}_T, \bar{y}_T) \in \mathcal{M} \times \mathcal{N}$ can be computed by $(\bar{x}_0, \bar{y}_0) = (0, 0)$ and the inductive formula: $\bar{x}_{t+1} = \mathrm{Exp}_{\bar{x}_t}(\frac{1}{t+1} \cdot \mathrm{Exp}_{\bar{x}_t}^{-1}(\hat{x}_t))$ and $\bar{y}_{t+1} = \mathrm{Exp}_{\bar{y}_t}(\frac{1}{t+1} \cdot \mathrm{Exp}_{\bar{y}_t}^{-1}(\hat{y}_t))$ for all $t = 0, 1, \ldots, T - 1$.*

**Remark 3.3** *Theorem 3.2 presents the last-iterate convergence rate of Algorithm 2 for solving geodesically strongly-convex-strongly-concave problems while Theorem 3.3 gives the time-average convergence rate when the function $f$ is only assumed to be geodesically convex-concave. Note that we carefully choose the stepsizes such that our upper bounds match the lower bounds established for stochastic min-max optimization problems in Euclidean spaces [79, 85, 86], in terms of the dependence on $\kappa$, $1/\epsilon$ and $\sigma^2$, up to log factors.*

**Discussions:** The last-iterate linear convergence rate in terms of Riemannian metrics is only limited to geodesically strongly convex-concave cases but other results, e.g., the average-iterate sublinear

convergence rate, are derived under more mild conditions. This is consistent with classical results in the Euclidean setting where geodesic convexity reduces to convexity; indeed, the last-iterate linear convergence rate in terms of squared Euclidean norm is only obtained for strongly convex-concave cases. As such, our setting is not restrictive. Moreover, Zhang et al. [1] showed that the existence of a global saddle point is only guaranteed under the geodesically convex-concave assumption. For geodesically nonconvex-concave or geodesically nonconvex-nonconcave cases, a global saddle point might not exist and new optimality notions are required before algorithmic design. This question remains open in the Euclidean setting and is beyond the scope of this paper. However, we remark that an interesting class of robustification problems are nonconvex-nonconcave min-max problems in the Euclidean setting can be geodesically convex-concave in the Riemannian setting; see Appendix B.

## 4 Experiments

We present numerical experiments on the task of robust principal component analysis (RPCA) for symmetric positive definite (SPD) matrices. In particular, we compare the performance of Algorithm 1 and 2 with different outputs, i.e., the last iterate $(x_T, y_T)$ versus the time-average iterate $(\bar{x}_T, \bar{y}_T)$ (see the precise definition in Theorem 3.3). Note that our implementations of both algorithms are based on the MANOPT package [87]. All the experiments were implemented in MATLAB R2021b on a workstation with a 2.6 GHz Intel Core i7 and 16GB of memory. Due to space limitations, some additional experimental results are deferred to Appendix G.

**Experimental setup.** The problem of RPCA [88, 89] can be formulated as the Riemannian min-max optimization problem with an SPD manifold and a sphere manifold. Formally, we have

$$\max_{M \in \mathcal{M}_{\text{PSD}}^d} \min_{x \in \mathcal{S}^d} \left\{ -x^\top M x - \frac{\alpha}{n} \sum_{i=1}^n d(M, M_i) \right\}. \tag{5}$$

In this formulation, $\alpha > 0$ denotes the penalty parameter, $\{M_i\}_{i \in [n]}$ is a sequence of given data SPD matrices, $\mathcal{M}_{\text{PSD}}^d = \{M \in \mathbb{R}^{d \times d} : M \succ 0, M = M^\top\}$ denotes the SPD manifold, $\mathcal{S}^d = \{x \in \mathbb{R}^d : \|x\| = 1\}$ denotes the sphere manifold and $d(\cdot, \cdot) : \mathcal{M}_{\text{PSD}}^d \times \mathcal{M}_{\text{PSD}}^d \mapsto \mathbb{R}$ is the Riemannian distance induced by the exponential map on the SPD manifold $\mathcal{M}_{\text{PSD}}^d$. As demonstrated by Zhang et al. [1], the problem of RPCA is nonconvex-nonconcave from a Euclidean perspective but is *locally geodesically strongly-convex-strongly-concave* and satisfies most of the assumptions that we make in this paper. In particular, the SPD manifold is complete with sectional curvature in $[-\frac{1}{2}, 1]$ [90] and the sphere manifold is complete with sectional curvature of $1$. Other reasons why we use such example are: (i) it is a classical one in ML; (ii) Zhang et al. [1] also uses this example and observes the linear convergence behavior; (iii) the numerical results show that the unicity of geodesics assumption may not be necessary in practice; and (iv) this is an application where both min and max sides are done on Riemannian manifolds.

Following the previous works of Zhang et al. [1] and Han et al. [78], we generate a sequence of data matrices $M_i$ satisfying that their eigenvalues are in the range of $[0.2, 4.5]$. In our experiment, we fix $\alpha = 1.0$ and also vary the problem dimension $d \in \{25, 50, 100\}$. The evaluation metric is set as gradient norm. We set $n = 40$ and $n = 200$ in Figure 1 and 2. For RCEG, we set $\eta = \frac{1}{2\ell}$ where $\ell > 0$ is selected via grid search. For SRCEG, we set $\eta_t = \min\{\frac{1}{2\ell}, \frac{a}{t}\}$ where $\ell, a > 0$ are selected via grid search. Additional results on the effect of stepsize are summarized in Appendix G.

**Experimental results.** Figure 1 summarizes the effects of different outputs for RCEG; indeed, RCEG-last and RCEG-avg refer to Algorithm 1 with last iterate and time-average iterate respectively. It is clear that the last iterate of RCEG consistently exhibits linear convergence to an optimal solution in all the settings, verifying our theoretical results in Theorem 3.1. In contrast, the average iterate of RCEG converges much slower than the last iterate of RCEG. The possible reason is that the problem of RPCA is *only* locally geodesically strongly-convex-strongly-concave and averaging with the iterates generated during early stage will significantly slow down the convergence of RCEG.

Figure 2 presents the comparison between SRCEG (with either last iterate or time-average iterate) and RCEG with last-iterate; here, SRCEG-last and SRCEG-avg refer to Algorithm 2 with last iterate and time-average iterate respectively. We observe that SRCEG with either last iterate or average iterate converge faster than RCEG at the early stage and all of them finally converge to an optimal solution. This demonstrates the effectiveness and efficiency of SRCEG in practice. It is also worth

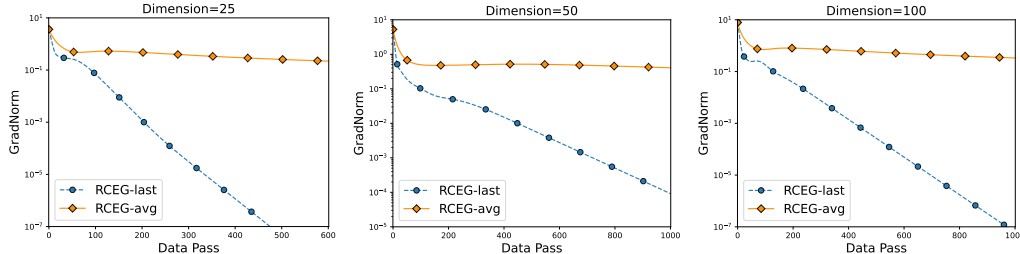

Figure 1: Comparison of last iterate (RCEG-last) and time-average iterate (RCEG-avg) for solving the RPCA problem in Eq. (5) with different problem dimensions $d \in \{25, 50, 100\}$. The horizontal axis represents the number of data passes and the vertical axis represents gradient norm.

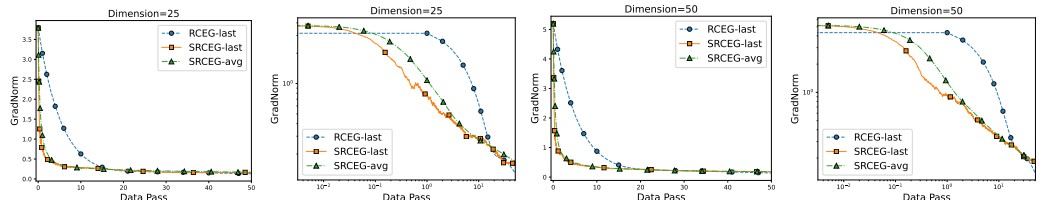

Figure 2: Comparison of RCEG and SRCEG for solving the RPCA problem in Eq. (5) with different problem dimensions $d \in \{25, 50\}$. The horizontal axis is the number of data passes and the vertical axis is gradient norm.

mentioning that the difference between last-iterate convergence and time-average-iterate convergence is not as significant as in the deterministic setting. This is possibly because the technique of averaging help cancels the negative effect of imperfect information [91, 92].

## 5    Conclusions

Inspired broadly by the structure of the complex competition that arises in many applications of robust optimization in ML, we focus on the problem of min-max optimization in the pure Riemannian setting (where both min and max player are constrained in a smooth manifold). Answering the open question of Zhang et al. [1] for the geodesically (strongly) convex-concave case, we showed that the Riemannian correction technique for EG matches the linear last-iterate complexity of their Euclidean counterparts in terms of accuracy and conditional number of objective for both deterministic and stochastic case. Additionally, we provide near-optimal guarantees for both smooth and non-smooth min-max optimization via Riemannian EG and GDA for the simple convex-concave case.

As a consequence of this work numerous open problems emerge; one immediate open question for future work is to explore whether the dependence on the curvature constant is also tight. Additionally, another generalization of interest would be to consider the performance of RCEG in the case of Riemannian Monotone Variational inequalities (RMVI) and examine the generalization of Zhang et al. [1] existence proof. Finally, there has been recent work in proving last-iterate convergence in the convex-concave setting via Sum-Of-Squares techniques [62]. It would be interesting to examine how one could leverage this machinery in a non-Euclidean but geodesic-metric-friendly framework.

### Acknowledgements

We would like to thank the area chair and five anonymous referees for constructive suggestions that improve the paper. This work was supported in part by the Mathematical Data Science program of the Office of Naval Research under grant number N00014-18-1-2764 and by the Vannevar Bush Faculty Fellowship program under grant number N00014-21-1-2941. The work of Michael I. Jordan is also partially supported by NSF Grant IIS-1901252. Emmanouil V. Vlatakis-Gkaragkounis is grateful for financial support by the Google-Simons Fellowship, Pancretan Association of America and Simons Collaboration on Algorithms and Geometry. This project was completed while he was a visiting research fellow at the Simons Institute for the Theory of Computing. Additionally, he would like to acknowledge the following series of NSF-CCF grants under the numbers 1763970/2107187/1563155/1814873.

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
