# Appendix

## Table of Contents

## A   Related Work

The literature for the geometric properties of Riemannian Manifolds is immense and hence we cannot hope to survey them here; for an appetizer, we refer the reader to Burago et al. [93] and Lee [94] and references therein. On the other hand, as stated, it is not until recently that the long-run non-asymptotic behavior of optimization algorithms in Riemannian manifolds (even the smooth ones) has encountered a lot of interest. For concision, we have deferred here a detailed exposition of the rest of recent results to Appendix A of the paper's supplement. Additionally, in Appendix B we also give a bunch of motivating examples which can be solved by Riemannian min-max optimization.

**Minimization on Riemannian manifolds.** Many application problems can be formulated as the minimization or maximization of a smooth function over Riemannian manifold and has triggered a line of research on the extension of the classical first-order and second-order methods to Riemannian setting with asymptotic convergence to first-order stationary points in general [95]. Recent years have witnessed the renewed interests on nonasymptotic convergence analysis of solution methods. In particular, Boumal et al. [96] proved the global sublinear convergence results for Riemannian gradient descent method and Riemannian trust region method, and further demonstrated that the Riemannian trust region method converges to a second-order stationary point in polynomial time; see also similar results in some other works [97–99]. We are also aware of recent works on problem-specific methods [100–102] and primal-dual methods [103].

Compared to the smooth counterpart, Riemannian nonsmooth optimization is harder and relatively less explored [104]. A few existing works focus on optimizing geodesically convex functions over Riemannian manifold with subgradient methods [83, 105, 106]. In particular, Ferreira and Oliveira [105] provided the first asymptotic convergence result while Zhang and Sra [83] and [106]

proved an nonasymptotic global convergence rate of $O(\epsilon^{-2})$ for Riemannian subgradient methods. Further, Ferreira and Oliveira [107] assumed that the proximal mapping over Riemannian manifold is computationally tractable and proved the global sublinear convergence of Riemannian proximal point method. Focusing on optimization over Stiefel manifold, Chen et al. [108] studied the composite objective function and proposed Riemannian proximal gradient method which only needs to compute the proximal mapping of nonsmooth component function over the tangent space of Stiefel manifold. Li et al. [109] consider optimizing a weakly convex function over Stiefel manifold and proposed Riemannian subgradient methods that drive a near-optimal stationarity measure below $\epsilon$ within the number of iterations bounded by $O(\epsilon^{-4})$.

There are some results on stochastic optimization over Riemannian manifold. In particular, Bonnabel [29] proved the first asymptotic convergence result for Riemannian stochastic gradient descent, which is extended by a line of subsequent works [110–113]. If the Riemannian Hessian is not positive definite, some recent works have suggested frameworks to escape saddle points [25, 114].

**Min-Max optimization in Euclidean spaces.** Focusing on solving specifically min-max problems, the algorithms under euclidean geometry have a very rich history in optimization that goes back at least to the original proximal point algorithms [115, 116] for variational inequality (VI) problems; At a high level, if the objective function is Lipschitz and strictly convex-concave, the simple forward-backward schemes are known to converge – and if combined with a Polyak–Ruppert averaging scheme [117–119], they achieve an $O(1/\epsilon^2)$ complexity[2] without the caveat of strictness [120]. If, in addition, the objective admits Lipschitz continuous gradients, then the extragradient (EG) algorithm [121] achieves trajectory convergence without strict monotonicity requirements, while the time-average iterate converges at $O(1/\epsilon)$ steps [122]. Finally, if the problem is strongly convex-concave, forward-backward methods computes an $\epsilon$-saddle point at $O(1/\epsilon)$ steps; and if the operator is also Lipschitz continuous, classical results in operator theory show that simple forward-backward methods suffice to achieve a linear convergence rate [120, 123].

**Min-Max optimization on Riemannian manifolds.** In the case of nonlinear geometry, the literature has been devoted on two different orthogonal axes: *a)* the existence of saddle point for min-max objective bi-functions and *b)* the design of algorithms for the computation of such points. For the existence of saddle point, a long line of recent work tried to generalize the seminal minima theorem for quasi-convex-quasi-concave problems of Sion [69]. The crucial bottleneck of this generalization to Riemannian smooth manifolds had been the application of both Knaster–Kuratowski–Mazurkiewicz (KKM) theorem and Helly's theorem in non-flat spaces. Before Zhang et al. [1], the existence of saddle points had been identified for the special case of Hadamard manifolds [73–75, 106].

Similar with the existence results, initially the developed methods referred to the computation of singularities in monotone variational operators typically in hyperbolic Hadamard manifolds with negative curvature [124]. More recently, Huang et al. [77] proposed a Riemannian gradient descent ascent method (RGDA), yet the analysis is restricted to $\mathcal{N}$ being a convex subset of the Euclidean space and $f(x, y)$ being strongly concave in $y$. It is worth mentioning that for the case Hadamard and generally hyperbolic manifolds, extra-gradient style algorithms have been proposed [125, 126] in the literature, establishing mainly their asymptotic convergence. However it was not until recent Zhang et al. [1] that the riemannian correction trick has been analyzed for the case of the extra-gradient algorithm. Bearing in our mind the higher-order methods, Han et al. [78] has recently proposed the Riemannian Hamiltonian Descent and versions of Newton's method for for geodesic convex geodesic concave functions. Since in this work, we focus only on first-order methods, we don't compare with the aforementioned Hamiltonian alternative since it incorporates always the extra computational burden of second-derivatives and hessian over a manifold.

## B  Motivating Examples

We provide some examples of Riemannian min-max optimization to give a sense of their expressivity. Two of the examples are the generic models from the optimization literature [95, 127, 128] and the

---

[2]For the rest of the presentation, we adopt the convention of presenting the *fine-grained complexity* performance measure for computing an $O(\epsilon)$-close solution instead of the *convergence rate* of a method. Thus a rate of the form $\|\mathbf{x}_t - \mathbf{x}^*\| \leq O(1/t^{1/p})$ typically corresponds to $O(1/\epsilon^p)$ gradient computations and the geometric rate $\|\mathbf{x}_t - \mathbf{x}^*\| \leq O(\exp(-\mu t))$ matches usually up with the $O(\ln(1/\epsilon))$ computational complexity.

two others are the formulations of application problems arising from machine learning and data analytics [34, 129, 130].

**Example B.1 (Riemannian optimization with nonlinear constraints)** *We can consider a rather straightforward generalization of constrained optimization problem from Euclidean spaces to Riemannian manifolds [131]. This formulation finds a wide range of real-world applications, e.g., non-negative principle component analysis, weighted max-cut and so on. Letting $\mathcal{M}$ be a finite-dimensional Riemannian manifold with unique geodesic, we focus on the following problem:*

$$\min_{x \in \mathcal{M}} f(x), \quad \text{s.t. } g(x) \leq 0, \ h(x) = 0,$$

*where $g := (g_1, g_2, \ldots, g_m) : \mathcal{M} \mapsto \mathbb{R}^m$ and $h := (h_1, h_2, \ldots, h_n) : \mathcal{M} \mapsto \mathbb{R}^n$ are two mappings. Then, we can introduce the dual variables $\lambda$ and $\mu$ and reformulate the aforementioned constrained optimization problem as follows,*

$$\min_{x \in \mathcal{M}} \max_{(\lambda, \mu) \in \mathbb{R}_+^m \times \mathbb{R}^n} f(x) + \langle \lambda, g(x) \rangle + \langle \mu, h(x) \rangle.$$

*Suppose that $f$ and all of $g_i$ and $h_i$ are geodesically convex and smooth, the above problem is a geodesic-convex-Euclidean-concave min-max optimization problem.*

**Example B.2 (Distributionally robust Riemannian optimization)** *Distributionally robust optimization (DRO) is an effective method to deal with the noisy data, adversarial data, and imbalanced data. We consider the problem of DRO over Riemannian manifold; indeed, given a set of data samples $\{\xi_i\}_{i=1}^N$, the problem of DRO over Riemannian manifold $\mathcal{M}$ can be written in the form of*

$$\min_{x \in \mathcal{M}} \max_{\boldsymbol{p} \in \mathcal{S}} \sum_{i=1}^N p_i \ell(x; \xi_i) - \|\boldsymbol{p} - \tfrac{1}{N}\boldsymbol{I}\|^2,$$

*where $\boldsymbol{p} = (p_1, p_2, \ldots, p_N)$ and $\mathcal{S} = \{\boldsymbol{p} \in \mathbb{R}^N : \sum_{i=1}^N p_i = 1, p_i \geq 0\}$. In general, $\ell(x; \xi_i)$ denotes the loss function over Riemannian manifold $\mathcal{M}$. If $\ell$ is geodesically convex and smooth, the above problem is a geodesic-convex-Euclidean-concave min-max optimization problem.*

**Example B.3 (Robust matrix Karcher mean problem)** *We consider a robust version of classical matrix Karcher mean problem. More specifically, the Karcher mean of $N$ symmetric positive definite matrices $\{A_i\}_{i=1}^N$ is defined as the matrix $X \in \mathcal{M} = \{X \in \mathbb{R}^{n \times n} : X \succ 0, \ X = X^\top\}$ that minimizes the sum of squared distance induced by the Riemannian metric:*

$$d(X, Y) = \|\log(X^{-1/2} Y X^{-1/2})\|_F.$$

*The loss function is thus defined by*

$$f(X; \{A_i\}_{i=1}^N) = \sum_{i=1}^N (d(X, A_i))^2.$$

*which is known to be nonconvex in Euclidean spaces but geodesically strongly convex. Then, the robust version of classical matrix Karcher mean problem is aiming at solving the following problem:*

$$\min_{X \in \mathcal{M}} \max_{Y_i \in \mathcal{M}} f(X; \{Y_i\}_{i=1}^N) - \gamma \left( \sum_{i=1}^N (d(Y_i, A_i))^2 \right),$$

*where $\gamma > 0$ stands for the trade-off between the computation of Karcher mean over a set of $\{Y_i\}_{i=1}^N$ and the difference between the observed samples $\{A_i\}_{i=1}^N$ and $\{Y_i\}_{i=1}^N$. It is clear that the above problem is a geodesically strongly-convex-strongly-concave min-max optimization problem.*

**Example B.4 (Projection robust optimal transport problem)** *We consider the projection robust optimal transport (OT) problem – a robust variant of the OT problem – that achieves superior sample complexity bound [132]. Let $\{x_1, x_2, \ldots, x_n\} \subseteq \mathbb{R}^d$ and $\{y_1, y_2, \ldots, y_n\} \subseteq \mathbb{R}^d$ denote sets of $n$ atoms, and let $(r_1, r_2, \ldots, r_n)$ and $(c_1, c_2, \ldots, c_n)$ denote weight vectors. We define discrete probability measures $\mu = \sum_{i=1}^n r_i \delta_{x_i}$ and $\nu = \sum_{j=1}^n c_j \delta_{y_j}$. In this setting, the computation of*

*the* $k$-*dimensional projection robust OT distance between* $\mu$ *and* $\nu$ *resorts to solving the following problem:*

$$\max_{U \in \text{St}(d,k)} \min_{\pi \in \Pi(\mu,\nu)} \sum_{i=1}^{n} \sum_{j=1}^{n} \pi_{i,j} \| U^\top x_i - U^\top y_j \|^2,$$

*where* $\text{St}(d,k) = \{U \in \mathbb{R}^{d \times k} \mid U^\top U = I_k\}$ *is a Stiefel manifold and* $\Pi(r,c) = \{\pi \in \mathbb{R}_+^{n \times n} \mid \sum_{j=1}^{n} \pi_{ij} = r_i, \sum_{i=1}^{n} \pi_{ij} = c_j\}$ *is a transportation polytope. It is worth mentioning that the above problem is a geodesically-nonconvex-Euclidean-concave min-max optimization problem with special structures, making the computation of stationary points tractable. While the global convergence guarantee for our algorithm does not apply, the above problem might be locally geodesically-convex-Euclidean-concave such that our algorithm with sufficiently good initialization works here.*

In addition to these examples, it is worth mentioning that Riemannian min-max optimization problems contain all general min-max optimization problems in Euclidean spaces and all Riemannian minimization or maximization optimization problems. It is also an abstraction of many machine learning problems, e.g,. principle component analysis [2], dictionary learning [3, 4], deep neural networks (DNNs) [5] and low-rank matrix learning [133, 134]; indeed, the problem of principle component analysis resorts to optimization problems on Grassmann manifolds for example.

## C Metric Geometry

To generalize the first-order methods in Euclidean setting, we introduce several basic concepts in metric geometry [93], which are known to include both Euclidean spaces and Riemannian manifolds as special cases. Formally, we have

**Definition C.1 (Metric Space)** *A metric space* $(X, d)$ *is a pair of a set* $X$ *and a distance function* $d(\cdot, \cdot)$ *satisfying: (i)* $d(x, x') \geq 0$ *for any* $x, x' \in X$; *(ii)* $d(x, x') = d(x', x)$ *for any* $x, x' \in X$; *and (iii)* $d(x, x'') \leq d(x, x') + d(x', x'')$ *for any* $x, x', x'' \in X$. *In other words, the distance function* $d(\cdot, \cdot)$ *is non-negative, symmetrical and satisfies the triangle inequality.*

A *path* $\gamma : [0, 1] \mapsto X$ is a continuous mapping from the interval $[0, 1]$ to $X$ and the *length* of $\gamma$ is defined as $\text{length}(\gamma) := \lim_{n \to +\infty} \sup_{0=t_0 < \ldots < t_n=1} \sum_{i=1}^{n} d(\gamma(t_{i-1}), \gamma(t_i))$. Note that the triangle inequality implies that $\sup_{0=t_0 < \ldots < t_n=1} \sum_{i=1}^{n} d(\gamma(t_{i-1}), \gamma(t_i))$ is nondecreasing. Then, the length of a path $\gamma$ is well defined since the limit is either $+\infty$ or a finite scalar. Moreover, for $\forall \epsilon > 0$, there exists $n \in \mathbb{N}$ and the partition $0 = t_0 < \ldots < t_n = 1$ of the interval $[0, 1]$ such that $\text{length}(\gamma) \leq \sum_{i=1}^{n} d(\gamma(t_{i-1}), \gamma(t_i)) + \epsilon$.

**Definition C.2 (Length Space)** *A metric space* $(X, d)$ *is a length space if, for any* $x, x' \in X$ *and* $\epsilon > 0$, *there exists a path* $\gamma : [0, 1] \mapsto X$ *connecting* $x$ *and* $x'$ *such that* $\text{length}(\gamma) \leq d(x, x') + \epsilon$.

We can see from Definition C.2 that a set of length spaces is strict subclass of metric spaces; indeed, for some $x, x' \in X$, there does not exist a path $\gamma$ such that its length can be approximated by $d(x, x')$ for some tolerance $\epsilon > 0$. In metric geometry, a *geodesic* is a path which is locally a distance minimizer everywhere. More precisely, a path $\gamma$ is a geodesic if there is a constant $\nu > 0$ such that for any $t \in [0, 1]$ there is a neighborhood $I$ of $[0, 1]$ such that,

$$d(\gamma(t_1), \gamma(t_2)) = \nu |t_1 - t_2|, \quad \text{for any } t_1, t_2 \in I.$$

Note that the above generalizes the notion of geodesic for Riemannian manifolds. Then, we are ready to introduce the geodesic space and uniquely geodesic space [135].

**Definition C.3** *A metric space* $(X, d)$ *is a geodesic space if, for any* $x, x' \in X$, *there exists a geodesic* $\gamma : [0, 1] \mapsto X$ *connecting* $x$ *and* $x'$. *Furthermore, it is called uniquely geodesic if the geodesic connecting* $x$ *and* $x'$ *is unique for any* $x, x' \in X$.

Trigonometric geometry in nonlinear spaces is intrinsically different from Euclidean space. In particular, we remark that the law of cosines in Euclidean space (with $\| \cdot \|$ as $\ell_2$-norm) is crucial for analyzing the convergence property of optimization algorithms, e.g.,

$$\|a\|^2 = \|b\|^2 + \|c\|^2 - 2bc \cos(A),$$

where $a$, $b$, $c$ are sides of *a geodesic triangle* in Euclidean space and $A$ is the angle between $b$ and $c$. However, such nice property does not hold for nonlinear spaces due to the lack of flat geometry, further motivating us to extend the law of cosines under nonlinear trigonometric geometry. That is to say, given a geodesic triangle in $X$ with sides $a$, $b$, $c$ where $A$ is the angle between $b$ and $c$, we hope to establish the relationship between $a^2$, $b^2$, $c^2$ and $2bc \cos(A)$ in nonlinear spaces; see the main context for the comparing inequalities.

Finally, we specify the definition of *section curvature* of Riemannian manifolds and clarify how such quantity affects the trigonometric comparison inequalities. More specifically, the sectional curvature is defined as the Gauss curvature of a 2-dimensional sub-manifold that are obtained from the image of a two-dimensional subspace of a tangent space after exponential mapping. It is worth mentioning that the above 2-dimensional sub-manifold is locally isometric to a 2-dimensional sphere, a Euclidean plane, and a hyperbolic plane with the same Gauss curvature if its sectional curvature is positive, zero and negative respectively. Then we are ready to summarize the existing trigonometric comparison inequalities for Riemannian manifold with bounded sectional curvatures. Note that the following two propositions are the full version of Proposition 2.1 and will be used in our subsequent proofs.

**Proposition C.1** *Suppose that $\mathcal{M}$ is a Riemannian manifold with sectional curvature that is upper bounded by $\kappa_{\max}$ and let $\Delta$ be a geodesic triangle in $\mathcal{M}$ with the side length $a$, $b$, $c$ and $A$ which is the angle between $b$ and $c$. If $\kappa_{\max} > 0$, we assume the diameter of $\mathcal{M}$ is bounded by $\frac{\pi}{\sqrt{\kappa_{\max}}}$. Then, we have*

$$a^2 \geq \underline{\xi}(\kappa_{\max}, c) \cdot b^2 + c^2 - 2bc \cos(A),$$

*where $\underline{\xi}(\kappa, c) := 1$ for $\kappa \leq 0$ and $\underline{\xi}(\kappa, c) := c\sqrt{\kappa} \cot(c\sqrt{\kappa}) < 1$ for $\kappa > 0$.*

**Proposition C.2** *Suppose that $\mathcal{M}$ is a Riemannian manifold with sectional curvature that is lower bounded by $\kappa_{\min}$ and let $\Delta$ be a geodesic triangle in $\mathcal{M}$ with the side length $a$, $b$, $c$ and $A$ which is the angle between $b$ and $c$. Then, we have*

$$a^2 \leq \overline{\xi}(\kappa_{\min}, c) \cdot b^2 + c^2 - 2bc \cos(A),$$

*where $\overline{\xi}(\kappa, c) := c\sqrt{-\kappa} \coth(c\sqrt{-\kappa}) > 1$ if $\kappa < 0$ and $\overline{\xi}(\kappa, c) := 1$ if $\kappa \geq 0$.*

**Remark C.1** *Proposition C.1 and C.2 are simply the restatement of Alimisis et al. [82, Corollary 2.1] and Zhang and Sra [83, Lemma 5]. The former inequality is obtained when the sectional curvature is bounded from above while the latter inequality characterizes the relationship between the trigonometric lengths when the sectional curvature is bounded from below. If $\kappa_{\min} = \kappa_{\max} = 0$ (i.e., Euclidean spaces), we have $\overline{\xi}(\kappa_{\min}, c) = \underline{\xi}(\kappa_{\max}, c) = 1$. The proof is based on Toponogov's theorem and Riccati comparison estimate [136, Proposition 25] and we refer the interested readers to Zhang and Sra [83] and Alimisis et al. [82] for the details.*

# D   Riemannian Gradient Descent Ascent for Nonsmooth Setting

In this section, we propose and analyze Riemannian gradient descent ascent (RGDA) method for nonsmooth Riemannian min-max optimization and extend it to stochastic RGDA. We present our results on the optimal last-iterate convergence guarantee for geodesically strongly-convex-strongly-concave setting (both deterministic and stochastic) and time-average convergence guarantee for geodesically convex-concave setting (both deterministic and stochastic).

## D.1   Algorithmic scheme

Compared to Riemannian corrected extragradient (RCEG) method, our Riemannian gradient descent ascent (RGDA) method is a relatively straightforward generalization of GDA in Euclidean spaces. More specifically, we start with the scheme of GDA as follows (just consider $\mathcal{M}$ and $\mathcal{N}$ as convex constraint sets in Euclidean spaces),

$$x_{t+1} \quad \leftarrow \quad \text{proj}_{\mathcal{M}}(x_t - \eta_t \cdot g_x^t), \qquad y_{t+1} \quad \leftarrow \quad \text{proj}_{\mathcal{N}}(y_t + \eta_t \cdot g_y^t). \tag{6}$$

where $(g_x^t, g_y^t) \in (\partial_x f(x_t, y_t), \partial_y f(x_t, y_t))$ is one subgradient of $f$. By replacing the projection operator by the corresponding exponential map and the gradient by the corresponding Riemannian gradient, we have

$$x_{t+1} \leftarrow \text{Exp}_{x_t}(-\eta_t \cdot g_x^t), \quad y_{t+1} \leftarrow \text{Exp}_{y_t}(\eta_t \cdot g_y^t).$$

| **Algorithm 3** RGDA | **Algorithm 4** SRGDA |
|---|---|
| **Input:** initial points $(x_0, y_0)$ and stepsizes $\eta_t > 0$.
**for** $t = 0, 1, 2, \ldots, T-1$ **do**
    Query $(g_x^t, g_y^t) \leftarrow$ (subgrad$_x f(x_t, y_t)$, subgrad$_y f(x_t, y_t)$) as Riemannian subgradient of $f$ at a point $(x_t, y_t)$.
    $x_{t+1} \leftarrow \mathrm{Exp}_{x_t}(-\eta_t \cdot g_x^t)$.
    $y_{t+1} \leftarrow \mathrm{Exp}_{y_t}(\eta_t \cdot g_y^t)$.
**end for** | **Input:** initial points $(x_0, y_0)$ and stepsizes $\eta_t > 0$.
**for** $t = 0, 1, 2, \ldots, T-1$ **do**
    Query $(g_x^t, g_y^t)$ as a **noisy** estimator of Riemannian subgradient of $f$ at a point $(x_t, y_t)$.
    $x_{t+1} \leftarrow \mathrm{Exp}_{x_t}(-\eta_t \cdot g_x^t)$.
    $y_{t+1} \leftarrow \mathrm{Exp}_{y_t}(\eta_t \cdot g_y^t)$.
**end for** |

where $(g_x^t, g_y^t) \leftarrow$ (subgrad$_x f(x_t, y_t)$, subgrad$_y f(x_t, y_t)$) is one Riemannian subgradient of $f$. Then, we summarize the resulting scheme of RGDA method in Algorithm 3 and its stochastic extension with noisy estimators of Riemannian gradients of $f$ in Algorithm 4.

### D.2    Main results

We present our main results on the global convergence rate estimation for Algorithm 3 and 4 in terms of Riemannian gradient and noisy Riemannian gradient evaluations. The following assumptions are made throughout for geodesically strongly-convex-strongly-concave and geodesically convex-concave settings.

**Assumption D.1** *The objective function $f : \mathcal{M} \times \mathcal{N} \mapsto \mathbb{R}$ and manifolds $\mathcal{M}$ and $\mathcal{N}$ satisfy*

1. *$f$ is geodesically $L$-Lipschitz and geodesically strongly-convex-strongly-concave with $\mu > 0$.*
2. *The diameter of the domain $\{(x, y) \in \mathcal{M} \times \mathcal{N} : -\infty < f(x, y) < +\infty\}$ is bounded by $D > 0$.*
3. *The sectional curvatures of $\mathcal{M}$ and $\mathcal{N}$ are both bounded in the range $[\kappa_{\min}, +\infty)$ with $\kappa_{\min} \leq 0$.*

**Assumption D.2** *The objective function $f : \mathcal{M} \times \mathcal{N} \mapsto \mathbb{R}$ and manifolds $\mathcal{M}$ and $\mathcal{N}$ satisfy*

1. *$f$ is geodesically $L$-Lipschitz and geodesically convex-concave.*
2. *The diameter of the domain $\{(x, y) \in \mathcal{M} \times \mathcal{N} : -\infty < f(x, y) < +\infty\}$ is bounded by $D > 0$.*
3. *The sectional curvatures of $\mathcal{M}$ and $\mathcal{N}$ are both bounded in the range $[\kappa_{\min}, +\infty)$ with $\kappa_{\min} \leq 0$.*

Imposing the geodesically Lipschitzness condition is crucial to achieve finite-time convergence guarantee if we do not assume the geodesically smoothness condition. Note that we only require the lower bound for the sectional curvatures of manifolds and this is weaker than that presented in the main context.

Letting $(x^\star, y^\star) \in \mathcal{M} \times \mathcal{N}$ be a global saddle point of $f$ (it exists under either Assumption D.1 or D.2), we let $D_0 = (d_\mathcal{M}(x_0, x^\star))^2 + (d_\mathcal{N}(y_0, y^\star))^2 > 0$ and summarize our results for Algorithm 3 in the following theorems.

**Theorem D.1** *Under Assumption D.1 and let $\eta_t > 0$ satisfies that $\eta_t = \frac{1}{\mu} \min\{1, \frac{2}{t}\}$. There exists some $T > 0$ such that the output of Algorithm 3 satisfies that $(d(x_T, x^\star))^2 + (d(y_T, y^\star))^2 \leq \epsilon$ and the total number of Riemannian subgradient evaluations is bounded by*

$$O\left(\frac{\overline{\xi}_0 L^2}{\mu^2 \epsilon}\right),$$

*where $\overline{\xi}_0 = \overline{\xi}(\kappa_{\min}, D)$ measures the lower bound for the change of non-flatness in $\mathcal{M}$ and $\mathcal{N}$.*

**Theorem D.2** *Under Assumption D.2 and let $\eta_t > 0$ satisfies that $\eta_t = \frac{1}{L}\sqrt{\frac{D_0}{2\overline{\xi}_0 T}}$. There exists some $T > 0$ such that the output of Algorithm 3 satisfies that $f(\bar{x}_T, y^\star) - f(x^\star, \bar{y}_T) \leq \epsilon$ and the total number of Riemannian subgradient evaluations is bounded by*

$$O\left(\frac{\overline{\xi}_0 L^2 D_0}{\epsilon^2}\right),$$

*where $\overline{\xi}_0 = \overline{\xi}(\kappa_{\min}, D)$ measures the lower bound for the change of non-flatness in $\mathcal{M}$ and $\mathcal{N}$, and the time-average iterates $(\bar{x}_T, \bar{y}_T) \in \mathcal{M} \times \mathcal{N}$ can be computed by $(\bar{x}_0, \bar{y}_0) = (0,0)$ and the inductive formula: $\bar{x}_{t+1} = \mathrm{Exp}_{\bar{x}_t}(\frac{1}{t+1} \cdot \mathrm{Exp}_{\bar{x}_t}^{-1}(x_t))$ and $\bar{y}_{t+1} = \mathrm{Exp}_{\bar{y}_t}(\frac{1}{t+1} \cdot \mathrm{Exp}_{\bar{y}_t}^{-1}(y_t))$ for all $t = 0, 1, \dots, T-1$.*

**Remark D.1** *Theorem D.1 and D.2 establish the last-iterate and time-average rates of convergence of Algorithm 3 for solving Riemannian min-max optimization problems under Assumption D.1 and D.2 respectively. Further, the dependence on $L$ and $1/\epsilon$ can not be improved since it has matched the lower bound established for the nonsmooth min-max optimization problems in Euclidean spaces.*

In the scheme of SRGDA, we highlight that $(g_x^t, g_y^t)$ is a noisy estimators of Riemannian subgradient of $f$ at $(x_t, y_t)$. It is necessary to impose the conditions such that these estimators are unbiased and has bounded variance. By abuse of notation, we assume that

$$g_x^t = \mathrm{subgrad}_x f(x_t, y_t) + \xi_x^t, \qquad g_y^t = \mathrm{subgrad}_y f(x_t, y_t) + \xi_y^t, \tag{7}$$

where the noises $(\xi_x^t, \xi_y^t)$ satisfy that

$$\mathbb{E}[\xi_x^t] = 0, \qquad \mathbb{E}[\xi_y^t] = 0, \qquad \mathbb{E}[\|\xi_x^t\|^2 + \|\xi_y^t\|^2] \le \sigma^2. \tag{8}$$

We are ready to summarize our results for Algorithm 4 in the following theorems.

**Theorem D.3** *Under Assumption D.1 and let Eq. (7) and Eq. (8) hold with $\sigma > 0$ and let $\eta_t > 0$ satisfies that $\eta_t = \frac{1}{\mu} \min\{1, \frac{2}{t}\}$. There exists some $T > 0$ such that the output of Algorithm 4 satisfies that $\mathbb{E}[(d(x_T, x^\star))^2 + (d(y_T, y^\star))^2] \le \epsilon$ and the total number of noisy Riemannian gradient evaluations is bounded by*

$$O\left(\frac{\overline{\xi}_0(L^2 + \sigma^2)}{\mu^2 \epsilon}\right),$$

*where $\overline{\xi}_0 = \overline{\xi}(\kappa_{\min}, D)$ measures the lower bound for the change of non-flatness in $\mathcal{M}$ and $\mathcal{N}$.*

**Theorem D.4** *Under Assumption D.2 and let Eq. (7) and Eq. (8) hold with $\sigma > 0$ and let $\eta_t > 0$ satisfies that $\eta_t = \frac{1}{2} \sqrt{\frac{D_0}{\overline{\xi}_0(L^2 + \sigma^2)T}}$. There exists some $T > 0$ such that the output of Algorithm 4 satisfies that $\mathbb{E}[f(\bar{x}_T, y^\star) - f(x^\star, \bar{y}_T)] \le \epsilon$ and the total number of noisy Riemannian gradient evaluations is bounded by*

$$O\left(\frac{\overline{\xi}_0(L^2 + \sigma^2)D_0}{\epsilon^2}\right),$$

*where $\overline{\xi}_0 = \overline{\xi}(\kappa_{\min}, D)$ measures the lower bound for the change of non-flatness in $\mathcal{M}$ and $\mathcal{N}$, and the time-average iterates $(\bar{x}_T, \bar{y}_T) \in \mathcal{M} \times \mathcal{N}$ can be computed by $(\bar{x}_0, \bar{y}_0) = (0,0)$ and the inductive formula: $\bar{x}_{t+1} = \mathrm{Exp}_{\bar{x}_t}(\frac{1}{t+1} \cdot \mathrm{Exp}_{\bar{x}_t}^{-1}(x_t))$ and $\bar{y}_{t+1} = \mathrm{Exp}_{\bar{y}_t}(\frac{1}{t+1} \cdot \mathrm{Exp}_{\bar{y}_t}^{-1}(y_t))$ for all $t = 0, 1, \dots, T-1$.*

**Remark D.2** *Theorem D.3 and D.4 establish the last-iterate and time-average rates of convergence of Algorithm 4 for solving Riemannian min-max optimization problems under Assumption D.1 and D.2. Moreover, the dependence on $L$ and $1/\epsilon$ can not be improved since it has matched the lower bound established for nonsmooth stochastic min-max optimization problems in Euclidean spaces.*

# E  Missing Proofs for Riemannian Corrected Extragradient Method

In this section, we present some technical lemmas for analyzing the convergence property of Algorithm 1 and 2. We also give the proofs of Theorem 3.1, 3.2 and 3.3.

## E.1  Technical lemmas

We provide two technical lemmas for analyzing Algorithm 1 and 2 respectively. Parts of the first lemma were presented in Zhang et al. [1, Lemma C.1]. For the completeness, we provide the proof details.

**Lemma E.1** *Under Assumption 3.1 and let $\{(x_t, y_t), (\hat{x}_t, \hat{y}_t)\}_{t=0}^{T-1}$ be generated by Algorithm 1 with the stepsize $\eta > 0$. Then, we have*

$$0 \leq \tfrac{1}{2}\left((d_{\mathcal{M}}(x_t, x^\star))^2 - (d_{\mathcal{M}}(x_{t+1}, x^\star))^2 + (d_{\mathcal{N}}(y_t, y^\star))^2 - (d_{\mathcal{N}}(y_{t+1}, y^\star))^2\right)$$
$$+2\overline{\xi}_0\eta^2\ell^2((d_{\mathcal{M}}(\hat{x}_t, x_t))^2 + (d_{\mathcal{N}}(\hat{y}_t, y_t))^2) - \tfrac{1}{2}\underline{\xi}_0\left((d_{\mathcal{M}}(\hat{x}_t, x_t))^2 + (d_{\mathcal{N}}(\hat{y}_t, y_t))^2\right)$$
$$-\tfrac{\mu\eta}{2}\left((d_{\mathcal{M}}(\hat{x}_t, x^\star))^2 + (d_{\mathcal{N}}(\hat{y}_t, y^\star))^2\right).$$

*where $(x^\star, y^\star) \in \mathcal{M} \times \mathcal{N}$ is a global saddle point of $f$.*

*Proof.* Since $f$ is geodesically $\ell$-smooth, we have the Riemannian gradients of $f$, i.e., $(\mathrm{grad}_x f, \mathrm{grad}_y f)$, are well defined. Since $f$ is geodesically strongly-concave-strongly-concave with the modulus $\mu \geq 0$ (here $\mu = 0$ means that $f$ is geodesically concave-concave), we have

$$f(\hat{x}_t, y^\star) - f(x^\star, \hat{y}_t) = f(\hat{x}_t, \hat{y}_t) - f(x^\star, \hat{y}_t) - (f(\hat{x}_t, \hat{y}_t) - f(\hat{x}_t, y^\star))$$

$$\overset{\text{Definition 2.2}}{\leq} -\langle \mathrm{grad}_x f(\hat{x}_t, \hat{y}_t), \mathrm{Exp}_{\hat{x}_t}^{-1}(x^\star)\rangle + \langle \mathrm{grad}_y f(\hat{x}_t, \hat{y}_t), \mathrm{Exp}_{\hat{y}_t}^{-1}(y^\star)\rangle - \tfrac{\mu}{2}(d_{\mathcal{M}}(\hat{x}_t, x^\star))^2 - \tfrac{\mu}{2}(d_{\mathcal{N}}(\hat{y}_t, y^\star))^2.$$

Since $(x^\star, y^\star) \in \mathcal{M} \times \mathcal{N}$ is a global saddle point of $f$, we have $f(\hat{x}_t, y^\star) - f(x^\star, \hat{y}_t) \geq 0$. Recalling also from the scheme of Algorithm 1 that we have

$$x_{t+1} \leftarrow \mathrm{Exp}_{\hat{x}_t}(-\eta \cdot \mathrm{grad}_x f(\hat{x}_t, \hat{y}_t) + \mathrm{Exp}_{\hat{x}_t}^{-1}(x_t)),$$
$$y_{t+1} \leftarrow \mathrm{Exp}_{\hat{y}_t}(\eta \cdot \mathrm{grad}_y f(\hat{x}_t, \hat{y}_t) + \mathrm{Exp}_{\hat{y}_t}^{-1}(y_t)).$$

By the definition of an exponential map, we have

$$\begin{aligned}\mathrm{Exp}_{\hat{x}_t}^{-1}(x_{t+1}) &= -\eta \cdot \mathrm{grad}_x f(\hat{x}_t, \hat{y}_t) + \mathrm{Exp}_{\hat{x}_t}^{-1}(x_t), \\ \mathrm{Exp}_{\hat{y}_t}^{-1}(y_{t+1}) &= \eta \cdot \mathrm{grad}_y f(\hat{x}_t, \hat{y}_t) + \mathrm{Exp}_{\hat{y}_t}^{-1}(y_t).\end{aligned} \quad (9)$$

This implies that

$$\begin{aligned}-\langle \mathrm{grad}_x f(\hat{x}_t, \hat{y}_t), \mathrm{Exp}_{\hat{x}_t}^{-1}(x^\star)\rangle &= \tfrac{1}{\eta}(\langle \mathrm{Exp}_{\hat{x}_t}^{-1}(x_{t+1}), \mathrm{Exp}_{\hat{x}_t}^{-1}(x^\star)\rangle - \langle \mathrm{Exp}_{\hat{x}_t}^{-1}(x_t), \mathrm{Exp}_{\hat{x}_t}^{-1}(x^\star)\rangle), \\ \langle \mathrm{grad}_y f(\hat{x}_t, \hat{y}_t), \mathrm{Exp}_{\hat{y}_t}^{-1}(y^\star)\rangle &= \tfrac{1}{\eta}(\langle \mathrm{Exp}_{\hat{y}_t}^{-1}(y_{t+1}), \mathrm{Exp}_{\hat{y}_t}^{-1}(y^\star)\rangle - \langle \mathrm{Exp}_{\hat{y}_t}^{-1}(y_t), \mathrm{Exp}_{\hat{y}_t}^{-1}(y^\star)\rangle).\end{aligned}$$

Putting these pieces together yields that

$$0 \leq \tfrac{1}{\eta}(\langle \mathrm{Exp}_{\hat{x}_t}^{-1}(x_{t+1}), \mathrm{Exp}_{\hat{x}_t}^{-1}(x^\star)\rangle - \langle \mathrm{Exp}_{\hat{x}_t}^{-1}(x_t), \mathrm{Exp}_{\hat{x}_t}^{-1}(x^\star)\rangle) - \tfrac{\mu}{2}(d_{\mathcal{M}}(\hat{x}_t, x^\star))^2$$
$$+\tfrac{1}{\eta}(\langle \mathrm{Exp}_{\hat{y}_t}^{-1}(y_{t+1}), \mathrm{Exp}_{\hat{y}_t}^{-1}(y^\star)\rangle - \langle \mathrm{Exp}_{\hat{y}_t}^{-1}(y_t), \mathrm{Exp}_{\hat{y}_t}^{-1}(y^\star)\rangle) - \tfrac{\mu}{2}(d_{\mathcal{N}}(\hat{y}_t, y^\star))^2.$$

Equivalently, we have

$$0 \leq \langle \mathrm{Exp}_{\hat{x}_t}^{-1}(x_{t+1}), \mathrm{Exp}_{\hat{x}_t}^{-1}(x^\star)\rangle - \langle \mathrm{Exp}_{\hat{x}_t}^{-1}(x_t), \mathrm{Exp}_{\hat{x}_t}^{-1}(x^\star)\rangle - \tfrac{\mu\eta}{2}(d_{\mathcal{M}}(\hat{x}_t, x^\star))^2 \quad (10)$$
$$+\langle \mathrm{Exp}_{\hat{y}_t}^{-1}(y_{t+1}), \mathrm{Exp}_{\hat{y}_t}^{-1}(y^\star)\rangle - \langle \mathrm{Exp}_{\hat{y}_t}^{-1}(y_t), \mathrm{Exp}_{\hat{y}_t}^{-1}(y^\star)\rangle - \tfrac{\mu\eta}{2}(d_{\mathcal{N}}(\hat{y}_t, y^\star))^2.$$

It suffices to bound the terms in the right-hand side of Eq. (10) by leveraging the celebrated comparison inequalities on Riemannian manifold with bounded sectional curvature (see Proposition C.1 and C.2). More specifically, we define the constants using $\overline{\xi}(\cdot, \cdot)$ and $\underline{\xi}(\cdot, \cdot)$ from Proposition C.1 and C.2 as follows,

$$\overline{\xi}_0 = \overline{\xi}(\kappa_{\min}, D), \qquad \underline{\xi}_0 = \underline{\xi}(\kappa_{\max}, D).$$

By Proposition C.1 and using that $\max\{d_{\mathcal{M}}(\hat{x}_t, x^\star), d_{\mathcal{N}}(\hat{y}_t, y^\star)\} \leq D$, we have

$$\begin{aligned}-\langle \mathrm{Exp}_{\hat{x}_t}^{-1}(x_t), \mathrm{Exp}_{\hat{x}_t}^{-1}(x^\star)\rangle &\leq -\tfrac{1}{2}\left(\underline{\xi}_0(d_{\mathcal{M}}(\hat{x}_t, x_t))^2 + (d_{\mathcal{M}}(\hat{x}_t, x^\star))^2 - (d_{\mathcal{M}}(x_t, x^\star))^2\right), \\ -\langle \mathrm{Exp}_{\hat{y}_t}^{-1}(y_t), \mathrm{Exp}_{\hat{y}_t}^{-1}(y^\star)\rangle &\leq -\tfrac{1}{2}\left(\underline{\xi}_0(d_{\mathcal{N}}(\hat{y}_t, y_t))^2 + (d_{\mathcal{N}}(\hat{y}_t, y^\star))^2 - (d_{\mathcal{N}}(y_t, y^\star))^2\right).\end{aligned}$$
$$(11)$$

By Proposition C.2 and using that $\max\{d_{\mathcal{M}}(\hat{x}_t, x^\star), d_{\mathcal{N}}(\hat{y}_t, y^\star)\} \leq D$, we have

$$\langle \mathrm{Exp}_{\hat{x}_t}^{-1}(x_{t+1}), \mathrm{Exp}_{\hat{x}_t}^{-1}(x^\star)\rangle \leq \tfrac{1}{2}\left(\overline{\xi}_0(d_{\mathcal{M}}(\hat{x}_t, x_{t+1}))^2 + (d_{\mathcal{M}}(\hat{x}_t, x^\star))^2 - (d_{\mathcal{M}}(x_{t+1}, x^\star))^2\right).$$

and

$$\langle \mathrm{Exp}_{\hat{y}_t}^{-1}(y_{t+1}), \mathrm{Exp}_{\hat{y}_t}^{-1}(y^\star)\rangle \leq \tfrac{1}{2}\left(\overline{\xi}_0(d_{\mathcal{N}}(\hat{y}_t, y_{t+1}))^2 + (d_{\mathcal{N}}(\hat{y}_t, y^\star))^2 - (d_{\mathcal{N}}(y_{t+1}, y^\star))^2\right).$$

By the definition of an exponential map and Riemannian metric, we have

$$
\begin{array}{rcl}
d_{\mathcal{M}}(\hat{x}_t, x_{t+1}) & = & \|\mathrm{Exp}_{\hat{x}_t}^{-1}(x_{t+1})\| \overset{\mathrm{Eq.~(9)}}{=} \|\eta \cdot \mathrm{grad}_x f(\hat{x}_t, \hat{y}_t) - \mathrm{Exp}_{\hat{x}_t}^{-1}(x_t)\|, \\
d_{\mathcal{N}}(\hat{y}_t, y_{t+1}) & = & \|\mathrm{Exp}_{\hat{y}_t}^{-1}(y_{t+1})\| \overset{\mathrm{Eq.~(9)}}{=} \|\eta \cdot \mathrm{grad}_y f(\hat{x}_t, \hat{y}_t) + \mathrm{Exp}_{\hat{y}_t}^{-1}(y_t)\|.
\end{array}
\tag{12}
$$

Further, we see from the scheme of Algorithm 1 that we have

$$
\begin{array}{rcl}
\hat{x}_t & \leftarrow & \mathrm{Exp}_{x_t}(-\eta \cdot \mathrm{grad}_x f(x_t, y_t)), \\
\hat{y}_t & \leftarrow & \mathrm{Exp}_{y_t}(\eta \cdot \mathrm{grad}_y f(x_t, y_t)).
\end{array}
$$

By the definition of an exponential map, we have

$$
\mathrm{Exp}_{x_t}^{-1}(\hat{x}_t) = -\eta \cdot \mathrm{grad}_x f(x_t, y_t), \qquad \mathrm{Exp}_{y_t}^{-1}(\hat{y}_t) = \eta \cdot \mathrm{grad}_y f(x_t, y_t).
$$

Using the definition of a parallel transport map and the above equations, we have

$$
\mathrm{Exp}_{\hat{x}_t}^{-1}(x_t) = \eta \cdot \Gamma_{x_t}^{\hat{x}_t} \mathrm{grad}_x f(x_t, y_t), \qquad \mathrm{Exp}_{\hat{y}_t}^{-1}(y_t) = -\eta \cdot \Gamma_{y_t}^{\hat{y}_t} \mathrm{grad}_y f(x_t, y_t)
$$

Since $f$ is geodesically $\ell$-smooth, we have

$$
\begin{array}{rcl}
\|\mathrm{grad}_x f(\hat{x}_t, \hat{y}_t) - \Gamma_{x_t}^{\hat{x}_t} \mathrm{grad}_x f(x_t, y_t)\| & \leq & \ell(d_{\mathcal{M}}(\hat{x}_t, x_t) + d_{\mathcal{N}}(\hat{y}_t, y_t)), \\
\|\mathrm{grad}_y f(\hat{x}_t, \hat{y}_t) - \Gamma_{y_t}^{\hat{y}_t} \mathrm{grad}_y f(x_t, y_t)\| & \leq & \ell(d_{\mathcal{M}}(\hat{x}_t, x_t) + d_{\mathcal{N}}(\hat{y}_t, y_t)).
\end{array}
$$

Plugging the above inequalities into Eq. (12) yields that

$$
\max\left\{ d_{\mathcal{M}}(\hat{x}_t, x_{t+1}), d_{\mathcal{N}}(\hat{y}_t, y_{t+1}) \right\} \leq \eta\ell(d_{\mathcal{M}}(\hat{x}_t, x_t) + d_{\mathcal{N}}(\hat{y}_t, y_t)).
$$

Therefore, we have

$$
\begin{array}{rcl}
\langle \mathrm{Exp}_{\hat{x}_t}^{-1}(x_{t+1}), \mathrm{Exp}_{\hat{x}_t}^{-1}(x^\star)\rangle & \leq & \frac{1}{2}\left( 2\overline{\xi}_0 \eta^2 \ell^2 ((d_{\mathcal{M}}(\hat{x}_t, x_t))^2 + (d_{\mathcal{N}}(\hat{y}_t, y_t))^2) + (d_{\mathcal{M}}(\hat{x}_t, x^\star))^2 - (d_{\mathcal{M}}(x_{t+1}, x^\star))^2 \right), \\
\langle \mathrm{Exp}_{\hat{y}_t}^{-1}(y_{t+1}), \mathrm{Exp}_{\hat{y}_t}^{-1}(y^\star)\rangle & \leq & \frac{1}{2}\left( 2\overline{\xi}_0 \eta^2 \ell^2 ((d_{\mathcal{M}}(\hat{x}_t, x_t))^2 + (d_{\mathcal{N}}(\hat{y}_t, y_t))^2) + (d_{\mathcal{N}}(\hat{y}_t, y^\star))^2 - (d_{\mathcal{N}}(y_{t+1}, y^\star))^2 \right).
\end{array}
$$

Plugging the above inequalities and Eq. (11) into Eq. (10) yields the desired inequality. $\qquad\square$

The second lemma gives another key inequality that is satisfied by the iterates generated by Algorithm 2.

**Lemma E.2** *Under Assumption 3.1 (or Assumption 3.2) and the noisy model (cf. Eq. (3) and (4)) and let $\{(x_t, y_t), (\hat{x}_t, \hat{y}_t)\}_{t=0}^{T-1}$ be generated by Algorithm 2 with the stepsize $\eta > 0$. Then, we have*

$$
\begin{aligned}
\mathbb{E}[f(\hat{x}_t, y^\star) - f(x^\star, \hat{y}_t)] \leq & \ \frac{1}{2\eta}\mathbb{E}\left[ (d_{\mathcal{M}}(x_t, x^\star))^2 - (d_{\mathcal{M}}(x_{t+1}, x^\star))^2 + (d_{\mathcal{N}}(y_t, y^\star))^2 - (d_{\mathcal{N}}(y_{t+1}, y^\star))^2 \right] \\
& + 6\overline{\xi}_0 \eta\ell^2 \mathbb{E}\left[ (d_{\mathcal{M}}(\hat{x}_t, x_t))^2 + (d_{\mathcal{N}}(\hat{y}_t, y_t))^2 \right] - \frac{1}{2\eta}\underline{\xi}_0 \mathbb{E}\left[ (d_{\mathcal{M}}(\hat{x}_t, x_t))^2 + (d_{\mathcal{N}}(\hat{y}_t, y_t))^2 \right] \\
& - \frac{\mu}{2}\mathbb{E}\left[ (d_{\mathcal{M}}(\hat{x}_t, x^\star))^2 + (d_{\mathcal{N}}(\hat{y}_t, y^\star))^2 \right] + 3\overline{\xi}_0 \eta\sigma^2,
\end{aligned}
$$

*where $(x^\star, y^\star) \in \mathcal{M} \times \mathcal{N}$ is a global saddle point of $f$.*

*Proof.* Using the same argument, we have ($\mu = 0$ refers to geodesically convex-concave case)

$$
\begin{aligned}
f(\hat{x}_t, y^\star) - f(x^\star, \hat{y}_t) &= f(\hat{x}_t, \hat{y}_t) - f(x^\star, \hat{y}_t) - (f(\hat{x}_t, \hat{y}_t) - f(\hat{x}_t, y^\star)) \\
&\leq -\langle \mathrm{grad}_x f(\hat{x}_t, \hat{y}_t), \mathrm{Exp}_{\hat{x}_t}^{-1}(x^\star)\rangle + \langle \mathrm{grad}_y f(\hat{x}_t, \hat{y}_t), \mathrm{Exp}_{\hat{y}_t}^{-1}(y^\star)\rangle - \frac{\mu}{2}(d_{\mathcal{M}}(\hat{x}_t, x^\star))^2 - \frac{\mu}{2}(d_{\mathcal{N}}(\hat{y}_t, y^\star))^2.
\end{aligned}
$$

Combining the arguments used in Lemma E.1 and the scheme of Algorithm 2, we have

$$
\begin{array}{rcl}
-\langle \hat{g}_x^t, \mathrm{Exp}_{\hat{x}_t}^{-1}(x^\star)\rangle & = & \frac{1}{\eta}(\langle \mathrm{Exp}_{\hat{x}_t}^{-1}(x_{t+1}), \mathrm{Exp}_{\hat{x}_t}^{-1}(x^\star)\rangle - \langle \mathrm{Exp}_{\hat{x}_t}^{-1}(x_t), \mathrm{Exp}_{\hat{x}_t}^{-1}(x^\star)\rangle), \\
\langle \hat{g}_y^t, \mathrm{Exp}_{\hat{y}_t}^{-1}(y^\star)\rangle & = & \frac{1}{\eta}(\langle \mathrm{Exp}_{\hat{y}_t}^{-1}(y_{t+1}), \mathrm{Exp}_{\hat{y}_t}^{-1}(y^\star)\rangle - \langle \mathrm{Exp}_{\hat{y}_t}^{-1}(y_t), \mathrm{Exp}_{\hat{y}_t}^{-1}(y^\star)\rangle).
\end{array}
$$

Putting these pieces together with Eq. (3) yields that

$$
\begin{aligned}
f(\hat{x}_t, y^\star) - f(x^\star, \hat{y}_t) \leq & \ \frac{1}{\eta}(\langle \mathrm{Exp}_{\hat{x}_t}^{-1}(x_{t+1}), \mathrm{Exp}_{\hat{x}_t}^{-1}(x^\star)\rangle - \langle \mathrm{Exp}_{\hat{x}_t}^{-1}(x_t), \mathrm{Exp}_{\hat{x}_t}^{-1}(x^\star)\rangle) \\
& + \frac{1}{\eta}(\langle \mathrm{Exp}_{\hat{y}_t}^{-1}(y_{t+1}), \mathrm{Exp}_{\hat{y}_t}^{-1}(y^\star)\rangle - \langle \mathrm{Exp}_{\hat{y}_t}^{-1}(y_t), \mathrm{Exp}_{\hat{y}_t}^{-1}(y^\star)\rangle) - \frac{\mu}{2}(d_{\mathcal{M}}(\hat{x}_t, x^\star))^2 - \frac{\mu}{2}(d_{\mathcal{N}}(\hat{y}_t, y^\star))^2 \\
& + \langle \hat{\xi}_x^t, \mathrm{Exp}_{\hat{x}_t}^{-1}(x^\star)\rangle - \langle \hat{\xi}_y^t, \mathrm{Exp}_{\hat{y}_t}^{-1}(y^\star)\rangle.
\end{aligned}
\tag{13}
$$

By the same argument as used in Lemma E.1, we have

$$-\langle \mathrm{Exp}_{\hat{x}_t}^{-1}(x_t), \mathrm{Exp}_{\hat{x}_t}^{-1}(x^\star)\rangle \leq -\tfrac{1}{2}\left(\underline{\xi}_0(d_{\mathcal{M}}(\hat{x}_t, x_t))^2 + (d_{\mathcal{M}}(\hat{x}_t, x^\star))^2 - (d_{\mathcal{M}}(x_t, x^\star))^2\right),$$
$$-\langle \mathrm{Exp}_{\hat{y}_t}^{-1}(y_t), \mathrm{Exp}_{\hat{y}_t}^{-1}(y^\star)\rangle \leq -\tfrac{1}{2}\left(\underline{\xi}_0(d_{\mathcal{N}}(\hat{y}_t, y_t))^2 + (d_{\mathcal{N}}(\hat{y}_t, y^\star))^2 - (d_{\mathcal{N}}(y_t, y^\star))^2\right),$$

(14)

and

$$\langle \mathrm{Exp}_{\hat{x}_t}^{-1}(x_{t+1}), \mathrm{Exp}_{\hat{x}_t}^{-1}(x^\star)\rangle \leq \tfrac{1}{2}\left(\overline{\xi}_0\eta^2\|\hat{g}_x^t - \Gamma_{x_t}^{\hat{x}_t}g_x^t\|^2 + (d_{\mathcal{M}}(\hat{x}_t, x^\star))^2 - (d_{\mathcal{M}}(x_{t+1}, x^\star))^2\right),$$
$$\langle \mathrm{Exp}_{\hat{y}_t}^{-1}(y_{t+1}), \mathrm{Exp}_{\hat{y}_t}^{-1}(y^\star)\rangle \leq \tfrac{1}{2}\left(\overline{\xi}_0\eta^2\|\hat{g}_y^t - \Gamma_{y_t}^{\hat{y}_t}g_y^t\|^2 + (d_{\mathcal{N}}(\hat{y}_t, y^\star))^2 - (d_{\mathcal{N}}(y_{t+1}, y^\star))^2\right).$$

Since $f$ is geodesically $\ell$-smooth and Eq. (3) holds, we have

$$\|\hat{g}_x^t - \Gamma_{x_t}^{\hat{x}_t}g_x^t\|^2 \leq 3\|\hat{\xi}_x^t\|^2 + 3\|\xi_x^t\|^2 + 6\ell^2(d_{\mathcal{M}}(\hat{x}_t, x_t))^2 + 6\ell^2(d_{\mathcal{N}}(\hat{y}_t, y_t))^2,$$
$$\|\hat{g}_y^t - \Gamma_{y_t}^{\hat{y}_t}g_y^t\|^2 \leq 3\|\hat{\xi}_y^t\|^2 + 3\|\xi_y^t\|^2 + 6\ell^2(d_{\mathcal{M}}(\hat{x}_t, x_t))^2 + 6\ell^2(d_{\mathcal{N}}(\hat{y}_t, y_t))^2.$$

Therefore, we have

$$\langle \mathrm{Exp}_{\hat{x}_t}^{-1}(x_{t+1}), \mathrm{Exp}_{\hat{x}_t}^{-1}(x^\star)\rangle + \langle \mathrm{Exp}_{\hat{y}_t}^{-1}(y_{t+1}), \mathrm{Exp}_{\hat{y}_t}^{-1}(y^\star)\rangle$$
$$\leq 6\overline{\xi}_0\eta^2\ell^2((d_{\mathcal{M}}(\hat{x}_t, x_t))^2 + (d_{\mathcal{N}}(\hat{y}_t, y_t))^2) + \tfrac{3}{2}\overline{\xi}_0\eta^2(\|\hat{\xi}_x^t\|^2 + \|\xi_x^t\|^2 + \|\hat{\xi}_y^t\|^2 + \|\xi_y^t\|^2)$$
$$+ \tfrac{1}{2}\left((d_{\mathcal{M}}(\hat{x}_t, x^\star))^2 - (d_{\mathcal{M}}(x_{t+1}, x^\star))^2 + (d_{\mathcal{N}}(\hat{y}_t, y^\star))^2 - (d_{\mathcal{N}}(y_{t+1}, y^\star))^2\right).$$

Plugging the above inequalities and Eq. (14) into Eq. (13) yields that

$$f(\hat{x}_t, y^\star) - f(x^\star, \hat{y}_t) \leq \tfrac{1}{2\eta}\left((d_{\mathcal{M}}(x_t, x^\star))^2 - (d_{\mathcal{M}}(x_{t+1}, x^\star))^2 + (d_{\mathcal{N}}(y_t, y^\star))^2 - (d_{\mathcal{N}}(y_{t+1}, y^\star))^2\right)$$
$$+ 6\overline{\xi}_0\eta\ell^2((d_{\mathcal{M}}(\hat{x}_t, x_t))^2 + (d_{\mathcal{N}}(\hat{y}_t, y_t))^2) + \tfrac{3}{2}\overline{\xi}_0\eta(\|\hat{\xi}_x^t\|^2 + \|\xi_x^t\|^2 + \|\hat{\xi}_y^t\|^2 + \|\xi_y^t\|^2)$$
$$- \tfrac{1}{2\eta}\underline{\xi}_0\left((d_{\mathcal{M}}(\hat{x}_t, x_t))^2 + (d_{\mathcal{N}}(\hat{y}_t, y_t))^2\right) - \tfrac{\mu}{2}(d_{\mathcal{M}}(\hat{x}_t, x^\star))^2 - \tfrac{\mu}{2}(d_{\mathcal{N}}(\hat{y}_t, y^\star))^2$$
$$+ \langle \hat{\xi}_x^t, \mathrm{Exp}_{\hat{x}_t}^{-1}(x^\star)\rangle - \langle \hat{\xi}_y^t, \mathrm{Exp}_{\hat{y}_t}^{-1}(y^\star)\rangle.$$

Taking the expectation of both sides and using Eq. (4) yields the desired inequality. $\qquad\square$

### E.2 Proof of Theorem 3.1

Since Riemannian metrics satisfy the triangle inequality, we have

$$(d_{\mathcal{M}}(\hat{x}_t, x^\star))^2 + (d_{\mathcal{N}}(\hat{y}_t, y^\star))^2 \geq \tfrac{1}{2}((d_{\mathcal{M}}(x_t, x^\star))^2 + (d_{\mathcal{N}}(y_t, y^\star))^2) - (d_{\mathcal{M}}(\hat{x}_t, x_t))^2 + (d_{\mathcal{N}}(\hat{y}_t, y_t))^2.$$

Plugging the above inequality into the inequality from Lemma E.1 yields that

$$(d_{\mathcal{M}}(x_{t+1}, x^\star))^2 + (d_{\mathcal{N}}(y_{t+1}, y^\star))^2$$
$$\leq \left(1 - \tfrac{\mu\eta}{2}\right)\left((d_{\mathcal{M}}(x_t, x^\star))^2 + (d_{\mathcal{N}}(y_t, y^\star))^2\right) + (4\overline{\xi}_0\eta^2\ell^2 + \mu\eta - \underline{\xi}_0)((d_{\mathcal{M}}(\hat{x}_t, x_t))^2 + (d_{\mathcal{N}}(\hat{y}_t, y_t))^2).$$

Since $\eta = \min\{\frac{1}{4\ell\sqrt{\tau_0}}, \frac{\underline{\xi}_0}{2\mu}\}$, we have $4\overline{\xi}_0\eta^2\ell^2 + \mu\eta - \underline{\xi}_0 \leq 0$. By the definition, we have $\tau_0 \geq 1$, $\kappa \geq 1$ and $\underline{\xi}_0 \leq 1$. This implies that

$$1 - \tfrac{\mu\eta}{2} = 1 - \min\left\{\tfrac{1}{8\kappa\sqrt{\tau_0}}, \tfrac{\underline{\xi}_0}{4}\right\} > 0.$$

Putting these pieces together yields that

$$(d_{\mathcal{M}}(x_T, x^\star))^2 + (d_{\mathcal{N}}(y_T, y^\star))^2 \leq \left(1 - \min\left\{\tfrac{1}{8\kappa\sqrt{\tau_0}}, \tfrac{\underline{\xi}_0}{4}\right\}\right)^T (d_{\mathcal{M}}(x_0, x^\star))^2 + (d_{\mathcal{N}}(y_0, y^\star))^2$$
$$\leq \left(1 - \min\left\{\tfrac{1}{8\kappa\sqrt{\tau_0}}, \tfrac{\underline{\xi}_0}{4}\right\}\right)^T D_0.$$

This completes the proof.

### E.3 Proof of Theorem 3.2

Since Riemannian metrics satisfy the triangle inequality, we have

$(d_\mathcal{M}(\hat{x}_t, x^\star))^2 + (d_\mathcal{N}(\hat{y}_t, y^\star))^2 \geq \frac{1}{2}((d_\mathcal{M}(x_t, x^\star))^2 + (d_\mathcal{N}(y_t, y^\star))^2) - (d_\mathcal{M}(\hat{x}_t, x_t))^2 + (d_\mathcal{N}(\hat{y}_t, y_t))^2.$

Plugging the above inequality into the inequality from Lemma E.2 yields that

$$\mathbb{E}[f(\hat{x}_t, y^\star) - f(x^\star, \hat{y}_t)] \leq \frac{1}{2\eta}\mathbb{E}\left[(d_\mathcal{M}(x_t, x^\star))^2 - (d_\mathcal{M}(x_{t+1}, x^\star))^2 + (d_\mathcal{N}(y_t, y^\star))^2 - (d_\mathcal{N}(y_{t+1}, y^\star))^2\right]$$
$$+ (6\overline{\xi}_0\eta\ell^2 + \frac{\mu}{2} - \frac{1}{2\eta}\underline{\xi}_0)\mathbb{E}\left[(d_\mathcal{M}(\hat{x}_t, x_t))^2 + (d_\mathcal{N}(\hat{y}_t, y_t))^2\right] - \frac{\mu}{4}\mathbb{E}\left[(d_\mathcal{M}(\hat{x}_t, x^\star))^2 + (d_\mathcal{N}(\hat{y}_t, y^\star))^2\right] + 3\overline{\xi}_0\eta\sigma^2.$$

Since $(x^\star, y^\star) \in \mathcal{M} \times \mathcal{N}$ is a global saddle point of $f$, we have $\mathbb{E}[f(\hat{x}_t, y^\star) - f(x^\star, \hat{y}_t)] \geq 0$. Then, we have

$$\mathbb{E}\left[(d_\mathcal{M}(x_{t+1}, x^\star))^2 + (d_\mathcal{N}(y_{t+1}, y^\star))^2\right]$$
$$\leq \left(1 - \frac{\mu\eta}{2}\right)\mathbb{E}\left[(d_\mathcal{M}(x_t, x^\star))^2 + (d_\mathcal{N}(y_t, y^\star))^2\right] + (12\overline{\xi}_0\eta^2\ell^2 + \mu\eta - \underline{\xi}_0)\mathbb{E}\left[(d_\mathcal{M}(\hat{x}_t, x_t))^2 + (d_\mathcal{N}(\hat{y}_t, y_t))^2\right]$$
$$+ 6\overline{\xi}_0\eta^2\sigma^2.$$

Since $\eta \leq \min\{\frac{1}{24\ell\sqrt{\tau_0}}, \frac{\underline{\xi}_0}{2\mu}\}$, we have $12\overline{\xi}_0\eta^2\ell^2 + \mu\eta - \underline{\xi}_0 \leq 0$. This implies that

$$\mathbb{E}\left[(d_\mathcal{M}(x_{t+1}, x^\star))^2 + (d_\mathcal{N}(y_{t+1}, y^\star))^2\right] \leq \left(1 - \frac{\mu\eta}{2}\right)\mathbb{E}\left[(d_\mathcal{M}(x_t, x^\star))^2 + (d_\mathcal{N}(y_t, y^\star))^2\right] + 6\overline{\xi}_0\eta^2\sigma^2.$$

By the definition, we have $\tau_0 \geq 1$, $\kappa \geq 1$ and $\underline{\xi}_0 \leq 1$. This implies that

$$1 - \frac{\mu\eta}{2} \geq 1 - \min\left\{\frac{1}{48\kappa\sqrt{\tau_0}}, \frac{\underline{\xi}_0}{4}\right\} > 0.$$

By the inductive arguments, we have

$$\mathbb{E}\left[(d_\mathcal{M}(x_T, x^\star))^2 + (d_\mathcal{N}(y_T, y^\star))^2\right] \leq \left(1 - \frac{\mu\eta}{2}\right)^T\left((d_\mathcal{M}(x_0, x^\star))^2 + (d_\mathcal{N}(y_0, y^\star))^2\right) + 6\overline{\xi}_0\eta^2\sigma^2\left(\sum_{t=0}^{T-1}\left(1 - \frac{\mu\eta}{2}\right)^t\right)$$
$$\leq \left(1 - \frac{\mu\eta}{2}\right)^T D_0 + \frac{12\overline{\xi}_0\eta\sigma^2}{\mu}.$$

Since $\eta = \min\{\frac{1}{24\ell\sqrt{\tau_0}}, \frac{\underline{\xi}_0}{2\mu}, \frac{2(\log(T) + \log(\mu^2 D_0\sigma^{-2}))}{\mu T}\}$, we have

$$\left(1 - \frac{\mu\eta}{2}\right)^T D_0 \leq \left(1 - \min\left\{\frac{1}{48\kappa\sqrt{\tau_0}}, \frac{\underline{\xi}_0}{4}\right\}\right)^T D_0 + \left(1 - \frac{\log(\mu^2 D_0\sigma^{-2}T)}{T}\right)^T D_0$$
$$\overset{1+x\leq e^x}{\leq} \left(1 - \min\left\{\frac{1}{48\kappa\sqrt{\tau_0}}, \frac{\underline{\xi}_0}{4}\right\}\right)^T D_0 + \frac{\sigma^2}{\mu^2 T},$$

and

$$\frac{12\overline{\xi}_0\eta\sigma^2}{\mu} \leq \frac{24\overline{\xi}_0\sigma^2}{\mu^2 T}\log\left(\frac{\mu^2 D_0 T}{\sigma^2}\right).$$

Putting these pieces together yields that

$$\mathbb{E}\left[(d_\mathcal{M}(x_T, x^\star))^2 + (d_\mathcal{N}(y_T, y^\star))^2\right] \leq \left(1 - \min\left\{\frac{1}{48\kappa\sqrt{\tau_0}}, \frac{\underline{\xi}_0}{4}\right\}\right)^T D_0 + \frac{\sigma^2}{\mu^2 T} + \frac{24\overline{\xi}_0\sigma^2}{\mu^2 T}\log\left(\frac{\mu^2 D_0 T}{\sigma^2}\right).$$

This completes the proof.

### E.4 Proof of Theorem 3.3

By the inductive formulas of $\bar{x}_{t+1} = \text{Exp}_{\bar{x}_t}(\frac{1}{t+1} \cdot \text{Exp}_{\bar{x}_t}^{-1}(\hat{x}_t))$ and $\bar{y}_{t+1} = \text{Exp}_{\bar{y}_t}(\frac{1}{t+1} \cdot \text{Exp}_{\bar{y}_t}^{-1}(\hat{y}_t))$ and using Zhang et al. [1, Lemma C.2], we have

$$f(\bar{x}_T, y^\star) - f(x^\star, \bar{y}_T) \leq \frac{1}{T}\left(\sum_{t=0}^{T-1} f(\hat{x}_t, y^\star) - f(x^\star, \hat{y}_t)\right).$$

Plugging the above inequality into the inequality from Lemma E.2 yields that (recall that $\mu = 0$ in geodesically convex-concave setting here)

$$\mathbb{E}[f(\bar{x}_T, y^\star) - f(x^\star, \bar{y}_T)] \leq \frac{1}{2\eta T}\left((d_\mathcal{M}(x_0, x^\star))^2 + (d_\mathcal{N}(y_0, y^\star))^2\right)$$
$$+ \frac{1}{T}\left(6\overline{\xi}_0\eta\ell^2 - \frac{1}{2\eta}\underline{\xi}_0\right)\left(\sum_{t=0}^{T-1}\mathbb{E}\left[(d_\mathcal{M}(\hat{x}_t, x_t))^2 + (d_\mathcal{N}(\hat{y}_t, y_t))^2\right]\right) + 3\overline{\xi}_0\eta\sigma^2.$$

Since $\eta \leq \frac{1}{4\ell\sqrt{\tau_0}}$, we have $6\bar{\xi}_0\eta\ell^2 - \frac{1}{2\eta}\xi_0 \leq 0$. Then, this together with $(d_\mathcal{M}(x_0, x^\star))^2 + (d_\mathcal{N}(y_0, y^\star))^2 \leq D_0$ implies that

$$\mathbb{E}[f(\bar{x}_T, y^\star) - f(x^\star, \bar{y}_T)] \leq \frac{D_0}{2\eta T} + 3\bar{\xi}_0\eta\sigma^2.$$

Since $\eta = \min\{\frac{1}{4\ell\sqrt{\tau_0}}, \frac{1}{\sigma}\sqrt{\frac{D_0}{\bar{\xi}_0 T}}\}$, we have

$$\frac{D_0}{2\eta T} \leq \frac{2\ell D_0\sqrt{\tau_0}}{T} + \frac{\sigma}{2}\sqrt{\frac{\bar{\xi}_0 D_0}{T}},$$

and

$$3\bar{\xi}_0\eta\sigma^2 \leq 3\sigma\sqrt{\frac{\bar{\xi}_0 D_0}{T}}.$$

Putting these pieces together yields that

$$\mathbb{E}[f(\bar{x}_T, y^\star) - f(x^\star, \bar{y}_T)] \leq \frac{2\ell D_0\sqrt{\tau_0}}{T} + \frac{7\sigma}{2}\sqrt{\frac{\bar{\xi}_0 D_0}{T}}.$$

This completes the proof.

# F   Missing Proofs for Riemannian Gradient Descent Ascent

In this section, we present some technical lemmas for analyzing the convergence property of Algorithm 3 and 4. We also give the proofs of Theorem D.1, D.2, D.3 and D.4.

## F.1   Technical lemmas

We provide two technical lemmas for analyzing Algorithm 3 and 4 respectively. The first lemma gives a key inequality that is satisfied by the iterates generated by Algorithm 3.

**Lemma F.1** *Under Assumption D.1 (or Assumption D.2) and let $\{(x_t, y_t)\}_{t=0}^{T-1}$ be generated by Algorithm 3 with the stepsize $\eta_t > 0$. Then, we have*

$$f(x_t, y^\star) - f(x^\star, y_t) \leq \frac{1}{2\eta_t}\left((d_\mathcal{M}(x_t, x^\star))^2 - (d_\mathcal{M}(x_{t+1}, x^\star))^2\right)$$
$$+ \frac{1}{2\eta_t}\left((d_\mathcal{N}(y_t, y^\star))^2 - (d_\mathcal{N}(y_{t+1}, y^\star))^2\right) - \frac{\mu}{2}(d_\mathcal{M}(x_t, x^\star))^2 - \frac{\mu}{2}(d_\mathcal{N}(y_t, y^\star))^2 + \bar{\xi}_0\eta_t L^2,$$

*where $(x^\star, y^\star) \in \mathcal{M} \times \mathcal{N}$ is a global saddle point of $f$.*

*Proof.* Since $f$ is geodesically strongly-concave-strongly-concave with the modulus $\mu \geq 0$ (here $\mu = 0$ means that $f$ is geodesically concave-concave), we have

$$f(x_t, y^\star) - f(x^\star, y_t) = f(x_t, y_t) - f(x^\star, y_t) - (f(x_t, y_t) - f(x_t, y^\star))$$
$$\leq -\langle\text{subgrad}_x f(x_t, y_t), \text{Exp}_{x_t}^{-1}(x^\star)\rangle + \langle\text{subgrad}_y f(x_t, y_t), \text{Exp}_{y_t}^{-1}(y^\star)\rangle - \frac{\mu}{2}(d_\mathcal{M}(x_t, x^\star))^2 - \frac{\mu}{2}(d_\mathcal{N}(y_t, y^\star))^2.$$

Recalling also from the scheme of Algorithm 3 that we have

$$x_{t+1} \leftarrow \text{Exp}_{x_t}(-\eta_t \cdot \text{subgrad}_x f(x_t, y_t)),$$
$$y_{t+1} \leftarrow \text{Exp}_{y_t}(\eta_t \cdot \text{subgrad}_y f(x_t, y_t)).$$

By the definition of an exponential map, we have

$$\begin{aligned}\text{Exp}_{x_t}^{-1}(x_{t+1}) &= -\eta_t \cdot \text{subgrad}_x f(x_t, y_t), \\ \text{Exp}_{y_t}^{-1}(y_{t+1}) &= \eta_t \cdot \text{subgrad}_y f(x_t, y_t).\end{aligned} \tag{15}$$

This implies that

$$\begin{aligned}-\langle\text{subgrad}_x f(x_t, y_t), \text{Exp}_{x_t}^{-1}(x^\star)\rangle &= \frac{1}{\eta_t}\langle\text{Exp}_{x_t}^{-1}(x_{t+1}), \text{Exp}_{x_t}^{-1}(x^\star)\rangle, \\ \langle\text{subgrad}_y f(x_t, y_t), \text{Exp}_{y_t}^{-1}(y^\star)\rangle &= \frac{1}{\eta_t}\langle\text{Exp}_{y_t}^{-1}(y_{t+1}), \text{Exp}_{y_t}^{-1}(y^\star)\rangle.\end{aligned}$$

Putting these pieces together yields that

$$f(x_t, y^\star) - f(x^\star, y_t) \leq \frac{1}{\eta_t}\langle\text{Exp}_{x_t}^{-1}(x_{t+1}), \text{Exp}_{x_t}^{-1}(x^\star)\rangle \tag{16}$$
$$+ \frac{1}{\eta_t}\langle\text{Exp}_{y_t}^{-1}(y_{t+1}), \text{Exp}_{y_t}^{-1}(y^\star)\rangle - \frac{\mu}{2}(d_\mathcal{M}(x_t, x^\star))^2 - \frac{\mu}{2}(d_\mathcal{N}(y_t, y^\star))^2.$$

It suffices to bound the terms in the right-hand side of Eq. (16) by leveraging the celebrated comparison inequalities on Riemannian manifold with lower bounded sectional curvature (see Proposition C.2). More specifically, we define the constants using $\overline{\xi}(\cdot,\cdot)$ and $\underline{\xi}(\cdot,\cdot)$ from Proposition C.2 as follows,

$$\overline{\xi}_0 = \overline{\xi}(\kappa_{\min}, D).$$

By Proposition C.2 and using that $\max\{d_{\mathcal{M}}(x_t, x^\star), d_{\mathcal{N}}(y_t, y^\star)\} \leq D$, we have

$$
\begin{aligned}
\langle \mathrm{Exp}_{x_t}^{-1}(x_{t+1}), \mathrm{Exp}_{x_t}^{-1}(x^\star) \rangle &\leq \tfrac{1}{2}\left(\overline{\xi}_0(d_{\mathcal{M}}(x_t, x_{t+1}))^2 + (d_{\mathcal{M}}(x_t, x^\star))^2 - (d_{\mathcal{M}}(x_{t+1}, x^\star))^2\right), \\
\langle \mathrm{Exp}_{y_t}^{-1}(y_{t+1}), \mathrm{Exp}_{y_t}^{-1}(y^\star) \rangle &\leq \tfrac{1}{2}\left(\overline{\xi}_0(d_{\mathcal{N}}(y_t, y_{t+1}))^2 + (d_{\mathcal{N}}(y_t, y^\star))^2 - (d_{\mathcal{N}}(y_{t+1}, y^\star))^2\right).
\end{aligned}
$$

Since $f$ is geodesically $L$-Lipschitz, we have

$$\|\mathrm{subgrad}_x f(x_t, y_t)\| \leq L, \quad \|\mathrm{subgrad}_y f(x_t, y_t)\| \leq L.$$

By the definition of an exponential map and Riemannian metric, we have

$$
\begin{aligned}
d_{\mathcal{M}}(x_t, x_{t+1}) &= \|\mathrm{Exp}_{x_t}^{-1}(x_{t+1})\| \overset{\mathrm{Eq.\,(15)}}{=} \|\eta_t \cdot \mathrm{subgrad}_x f(x_t, y_t)\| \leq \eta_t L, \\
d_{\mathcal{N}}(y_t, y_{t+1}) &= \|\mathrm{Exp}_{y_t}^{-1}(y_{t+1})\| \overset{\mathrm{Eq.\,(15)}}{=} \|\eta_t \cdot \mathrm{subgrad}_y f(x_t, y_t)\| \leq \eta_t L.
\end{aligned}
$$

Putting these pieces together yields that

$$
\begin{aligned}
\langle \mathrm{Exp}_{x_t}^{-1}(x_{t+1}), \mathrm{Exp}_{x_t}^{-1}(x^\star) \rangle &\leq \tfrac{1}{2}\left(\overline{\xi}_0 \eta_t^2 L^2 + (d_{\mathcal{M}}(x_t, x^\star))^2 - (d_{\mathcal{M}}(x_{t+1}, x^\star))^2\right), \\
\langle \mathrm{Exp}_{y_t}^{-1}(y_{t+1}), \mathrm{Exp}_{y_t}^{-1}(y^\star) \rangle &\leq \tfrac{1}{2}\left(\overline{\xi}_0 \eta_t^2 L^2 + (d_{\mathcal{N}}(y_t, y^\star))^2 - (d_{\mathcal{N}}(y_{t+1}, y^\star))^2\right).
\end{aligned}
$$

Plugging the above inequalities into Eq. (16) yields the desired inequality. $\qquad\square$

The second lemma gives another key inequality that is satisfied by the iterates generated by Algorithm 4.

**Lemma F.2** *Under Assumption D.1 (or Assumption D.2) and the noisy model (cf. Eq. (7) and (8)) and let $\{(x_t, y_t)\}_{t=0}^{T-1}$ be generated by Algorithm 4 with the stepsize $\eta_t > 0$. Then, we have*

$$
\mathbb{E}[f(x_t, y^\star) - f(x^\star, y_t)] \leq \tfrac{1}{2\eta_t}\mathbb{E}\left[(d_{\mathcal{M}}(x_t, x^\star))^2 - (d_{\mathcal{M}}(x_{t+1}, x^\star))^2\right]
$$
$$
+ \tfrac{1}{2\eta_t}\mathbb{E}\left[(d_{\mathcal{N}}(y_t, y^\star))^2 - (d_{\mathcal{N}}(y_{t+1}, y^\star))^2\right] - \tfrac{\mu}{2}\mathbb{E}\left[(d_{\mathcal{M}}(x_t, x^\star))^2 + (d_{\mathcal{N}}(y_t, y^\star))^2\right] + 2\overline{\xi}_0 \eta_t (L^2 + \sigma^2),
$$

*where $(x^\star, y^\star) \in \mathcal{M} \times \mathcal{N}$ is a global saddle point of $f$.*

*Proof.* Using the same argument, we have ($\mu = 0$ refers to geodesically convex-concave case)

$$
f(x_t, y^\star) - f(x^\star, y_t) = f(x_t, y_t) - f(x^\star, y_t) - (f(x_t, y_t) - f(x_t, y^\star))
$$
$$
\leq -\langle \mathrm{subgrad}_x f(x_t, y_t), \mathrm{Exp}_{x_t}^{-1}(x^\star) \rangle + \langle \mathrm{subgrad}_y f(x_t, y_t), \mathrm{Exp}_{y_t}^{-1}(y^\star) \rangle - \tfrac{\mu}{2}(d_{\mathcal{M}}(x_t, x^\star))^2 - \tfrac{\mu}{2}(d_{\mathcal{N}}(y_t, y^\star))^2.
$$

Combining the arguments used in Lemma F.1 and the scheme of Algorithm 2, we have

$$
\begin{aligned}
-\langle g_x^t, \mathrm{Exp}_{x_t}^{-1}(x^\star) \rangle &= \tfrac{1}{\eta_t}\langle \mathrm{Exp}_{x_t}^{-1}(x_{t+1}), \mathrm{Exp}_{x_t}^{-1}(x^\star) \rangle, \\
\langle g_y^t, \mathrm{Exp}_{y_t}^{-1}(y^\star) \rangle &= \tfrac{1}{\eta_t}\langle \mathrm{Exp}_{y_t}^{-1}(y_{t+1}), \mathrm{Exp}_{y_t}^{-1}(y^\star) \rangle.
\end{aligned}
$$

Putting these pieces together with Eq. (7) yields that

$$
f(x_t, y^\star) - f(x^\star, y_t) \leq \tfrac{1}{\eta_t}\langle \mathrm{Exp}_{x_t}^{-1}(x_{t+1}), \mathrm{Exp}_{x_t}^{-1}(x^\star) \tag{17}
$$
$$
+ \tfrac{1}{\eta_t}\langle \mathrm{Exp}_{y_t}^{-1}(y_{t+1}), \mathrm{Exp}_{y_t}^{-1}(y^\star) \rangle - \tfrac{\mu}{2}(d_{\mathcal{M}}(x_t, x^\star))^2 - \tfrac{\mu}{2}(d_{\mathcal{N}}(y_t, y^\star))^2 + \langle \xi_x^t, \mathrm{Exp}_{x_t}^{-1}(x^\star) \rangle - \langle \xi_y^t, \mathrm{Exp}_{y_t}^{-1}(y^\star) \rangle.
$$

By the same argument as used in Lemma F.1 and Eq. (7), we have

$$
\begin{aligned}
\langle \mathrm{Exp}_{x_t}^{-1}(x_{t+1}), \mathrm{Exp}_{x_t}^{-1}(x^\star) \rangle &\leq \tfrac{1}{2}\left(\overline{\xi}_0(d_{\mathcal{M}}(x_t, x_{t+1}))^2 + (d_{\mathcal{M}}(x_t, x^\star))^2 - (d_{\mathcal{M}}(x_{t+1}, x^\star))^2\right), \\
\langle \mathrm{Exp}_{y_t}^{-1}(y_{t+1}), \mathrm{Exp}_{y_t}^{-1}(y^\star) \rangle &\leq \tfrac{1}{2}\left(\overline{\xi}_0(d_{\mathcal{N}}(y_t, y_{t+1}))^2 + (d_{\mathcal{N}}(y_t, y^\star))^2 - (d_{\mathcal{N}}(y_{t+1}, y^\star))^2\right),
\end{aligned}
$$

and

$$
\begin{aligned}
d_{\mathcal{M}}(x_t, x_{t+1}) &= \|\mathrm{Exp}_{x_t}^{-1}(x_{t+1})\| = \|\eta_t \cdot g_x^t\| \leq \eta_t(L + \|\xi_x^t\|), \\
d_{\mathcal{N}}(y_t, y_{t+1}) &= \|\mathrm{Exp}_{y_t}^{-1}(y_{t+1})\| = \|\eta_t \cdot g_y^t\| \leq \eta_t(L + \|\xi_y^t\|).
\end{aligned}
$$

Therefore, we have

$$\langle \mathrm{Exp}_{x_t}^{-1}(x_{t+1}), \mathrm{Exp}_{x_t}^{-1}(x^\star) \rangle + \langle \mathrm{Exp}_{y_t}^{-1}(y_{t+1}), \mathrm{Exp}_{y_t}^{-1}(y^\star) \rangle$$

$$\leq \tfrac{1}{2}\overline{\xi}_0 \eta_t^2 (4L^2 + 2\|\xi_x^t\|^2 + 2\|\xi_y^t\|^2) + \tfrac{1}{2}\left((d_\mathcal{M}(x_t, x^\star))^2 - (d_\mathcal{M}(x_{t+1}, x^\star))^2 + (d_\mathcal{N}(y_t, y^\star))^2 - (d_\mathcal{N}(y_{t+1}, y^\star))^2\right).$$

Plugging the above inequalities into Eq. (17) yields that

$$f(x_t, y^\star) - f(x^\star, y_t) \leq \tfrac{1}{2\eta_t}\left((d_\mathcal{M}(x_t, x^\star))^2 - (d_\mathcal{M}(x_{t+1}, x^\star))^2 + (d_\mathcal{N}(y_t, y^\star))^2 - (d_\mathcal{N}(y_{t+1}, y^\star))^2\right)$$

$$+\overline{\xi}_0 \eta_t (2L^2 + \|\xi_x^t\|^2 + \|\xi_y^t\|^2) - \tfrac{\mu}{2}(d_\mathcal{M}(x_t, x^\star))^2 - \tfrac{\mu}{2}(d_\mathcal{N}(y_t, y^\star))^2 + \langle \xi_x^t, \mathrm{Exp}_{x_t}^{-1}(x^\star) \rangle - \langle \xi_y^t, \mathrm{Exp}_{y_t}^{-1}(y^\star) \rangle.$$

Taking the expectation of both sides and using Eq. (8) yields the desired inequality. $\qquad\square$

## F.2   Proof of Theorem D.1

Since $(x^\star, y^\star) \in \mathcal{M} \times \mathcal{N}$ is a global saddle point of $f$, we have $f(x_t, y^\star) - f(x^\star, y_t) \geq 0$. Plugging this inequality into the inequality from Lemma F.1 yields that

$$(d_\mathcal{M}(x_{t+1}, x^\star))^2 + (d_\mathcal{N}(y_{t+1}, y^\star))^2 \leq (1 - \mu\eta_t)\left((d_\mathcal{M}(x_t, x^\star))^2 + (d_\mathcal{N}(y_t, y^\star))^2\right) + 2\overline{\xi}_0 \eta_t^2 L^2.$$

Since $\eta_t = \tfrac{1}{\mu}\min\{1, \tfrac{2}{t}\}$, we have

$$(d_\mathcal{M}(x_{t+1}, x^\star))^2 + (d_\mathcal{N}(y_{t+1}, y^\star))^2 \leq (1 - \tfrac{2}{t})\left((d_\mathcal{M}(x_t, x^\star))^2 + (d_\mathcal{N}(y_t, y^\star))^2\right) + \tfrac{8\overline{\xi}_0 L^2}{\mu^2 t^2}, \quad \text{for all } t \geq 2.$$

Letting $\{b_t\}_{t\geq 1}$ be a nonnegative sequence such that $a_{t+1} \leq (1 - \tfrac{P}{t})a_t + \tfrac{Q}{t^2}$ where $P > 1$ and $Q > 0$. Then, Chung [137] proved that $a_t \leq \tfrac{Q}{P-1}\tfrac{1}{t}$. Therefore, we have

$$(d_\mathcal{M}(x_t, x^\star))^2 + (d_\mathcal{N}(y_t, y^\star))^2 \leq \tfrac{8\overline{\xi}_0 L^2}{\mu^2 t}, \quad \text{for all } t \geq 2.$$

This completes the proof.

## F.3   Proof of Theorem D.2

By the inductive formulas of $\bar{x}_{t+1} = \mathrm{Exp}_{\bar{x}_t}(\tfrac{1}{t+1} \cdot \mathrm{Exp}_{\bar{x}_t}^{-1}(x_t))$ and $\bar{y}_{t+1} = \mathrm{Exp}_{\bar{y}_t}(\tfrac{1}{t+1} \cdot \mathrm{Exp}_{\bar{y}_t}^{-1}(y_t))$ and using Zhang et al. [1, Lemma C.2], we have

$$f(\bar{x}_T, y^\star) - f(x^\star, \bar{y}_T) \leq \tfrac{1}{T}\left(\sum_{t=0}^{T-1} f(x_t, y^\star) - f(x^\star, y_t)\right).$$

Plugging the above inequality into the inequality from Lemma F.1 yields that (recall that $\mu = 0$ in geodesically convex-concave setting and $\eta_t = \eta = \tfrac{1}{L}\sqrt{\tfrac{D_0}{2\overline{\xi}_0 T}}$)

$$f(\bar{x}_T, y^\star) - f(x^\star, \bar{y}_T) \leq \tfrac{1}{2\eta T}\left((d_\mathcal{M}(x_0, x^\star))^2 + (d_\mathcal{N}(y_0, y^\star))^2\right) + \overline{\xi}_0 \eta L^2.$$

This together with $(d_\mathcal{M}(x_0, x^\star))^2 + (d_\mathcal{N}(y_0, y^\star))^2 \leq D_0$ implies that

$$f(\bar{x}_T, y^\star) - f(x^\star, \bar{y}_T) \leq \tfrac{D_0}{2\eta T} + \overline{\xi}_0 \eta L^2.$$

Since $\eta = \tfrac{1}{L}\sqrt{\tfrac{D_0}{2\overline{\xi}_0 T}}$, we have

$$f(\bar{x}_T, y^\star) - f(x^\star, \bar{y}_T) \leq L\sqrt{\tfrac{2\overline{\xi}_0 D_0}{T}}.$$

This completes the proof.

## F.4   Proof of Theorem D.3

Since $(x^\star, y^\star) \in \mathcal{M} \times \mathcal{N}$ is a global saddle point of $f$, we have $\mathbb{E}[f(x_t, y^\star) - f(x^\star, y_t)] \geq 0$. Plugging this inequality into the inequality from Lemma F.2 yields that

$$\mathbb{E}\left[(d_\mathcal{M}(x_{t+1}, x^\star))^2 + (d_\mathcal{N}(y_{t+1}, y^\star))^2\right] \leq (1 - \mu\eta_t)\mathbb{E}\left[(d_\mathcal{M}(x_t, x^\star))^2 + (d_\mathcal{N}(y_t, y^\star))^2\right] + 4\overline{\xi}_0 \eta_t^2 (L^2 + \sigma^2).$$

Since $\eta_t = \tfrac{1}{\mu}\min\{1, \tfrac{2}{t}\}$, we have

$$\mathbb{E}\left[(d_\mathcal{M}(x_{t+1}, x^\star))^2 + (d_\mathcal{N}(y_{t+1}, y^\star))^2\right] \leq (1 - \tfrac{2}{t})\mathbb{E}\left[(d_\mathcal{M}(x_t, x^\star))^2 + (d_\mathcal{N}(y_t, y^\star))^2\right] + \tfrac{16\overline{\xi}_0(L^2+\sigma^2)}{\mu^2 t^2}, \quad \text{for all } t \geq 2.$$

Applying the same argument as used in Theorem D.1, we have

$$(d_\mathcal{M}(x_t, x^\star))^2 + (d_\mathcal{N}(y_t, y^\star))^2 \leq \tfrac{16\overline{\xi}_0(L^2+\sigma^2)}{\mu^2 t}, \quad \text{for all } t \geq 2.$$

This completes the proof.

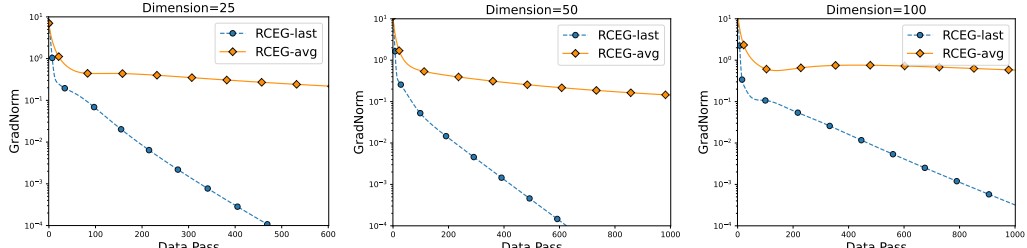

Figure 3: Comparison of last iterate (RCEG-last) and time-average iterate (RCEG-avg) for solving the RPCA problem when $\alpha = 2.0$. The horizontal axis represents the number of data passes and the vertical axis represents gradient norm.

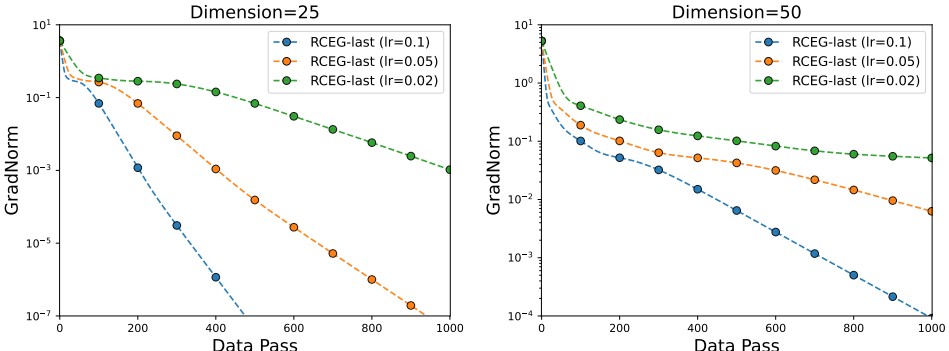

Figure 4: Comparison of different step sizes ($\eta \in \{0.1, 0.05, 0.02\}$) for solving the RPCA problem with different dimensions when $\alpha = 2.0$. The horizontal axis represents the number of data passes and the vertical axis represents gradient norm.

### F.5 Proof of Theorem D.4

Using the same argument, we have

$$f(\bar{x}_T, y^\star) - f(x^\star, \bar{y}_T) \le \tfrac{1}{T} \left( \sum_{t=0}^{T-1} f(x_t, y^\star) - f(x^\star, y_t) \right).$$

Plugging the above inequality into the inequality from Lemma F.2 yields that (recall that $\mu = 0$ in geodesically convex-concave setting and $\eta_t = \eta = \frac{1}{2}\sqrt{\frac{D_0}{\bar{\xi}_0(L^2+\sigma^2)T}}$)

$$\mathbb{E}[f(\bar{x}_T, y^\star) - f(x^\star, \bar{y}_T)] \le \tfrac{1}{2\eta T} \left( (d_{\mathcal{M}}(x_0, x^\star))^2 + (d_{\mathcal{N}}(y_0, y^\star))^2 \right) + 2\bar{\xi}_0 \eta (L^2 + \sigma^2).$$

This together with $(d_{\mathcal{M}}(x_0, x^\star))^2 + (d_{\mathcal{N}}(y_0, y^\star))^2 \le D_0$ implies that

$$\mathbb{E}[f(\bar{x}_T, y^\star) - f(x^\star, \bar{y}_T)] \le \tfrac{D_0}{2\eta T} + 2\bar{\xi}_0 \eta (L^2 + \sigma^2).$$

Since $\eta = \frac{1}{2}\sqrt{\frac{D_0}{\bar{\xi}_0(L^2+\sigma^2)T}}$, we have

$$f(\bar{x}_T, y^\star) - f(x^\star, \bar{y}_T) \le 2\sqrt{\tfrac{\bar{\xi}_0(L^2+\sigma^2)D_0}{T}}.$$

This completes the proof.

## G    Additional Experimental Results

We present some additional experimental results for the effect of different choices of $\alpha$ as well the effect of different choices of $\eta$ for for RCEG. In our experiment here, we set $n = 40$ consistently.

Figure 3 presents the performance of RCEG when $\alpha = 2.0$. We observe that the results are similar to that summarized in Figure 1. In particular, the last iterate of RCEG consistently achieves the linearly convergence to an optimal solution in all the settings. In contrast, the average iterate of RCEG converges much slower than the last iterate of RCEG. Figure 4 summarizes the effect of different choices of $\eta$ in RCEG. We observe that setting $\eta$ as a relatively larger value will speed up the convergence to an optimal solution while all of the choices here lead to the linear convergence. This suggests that the choice of stepsize $\eta$ in RCEG can be aggressive in practice.