# OpenReview forum: "First-Order Algorithms for Min-Max Optimization in Geodesic Metric Spaces"
_NeurIPS.cc/2022/Conference — NeurIPS 2022 Accept_

### Official Review · Reviewer_LyY4 · 2022-07-04

**Rating:** 6
**Confidence:** 3
**Soundness:** 3 good
**Presentation:** 3 good
**Contribution:** 2 fair

**Summary:**

The paper provides optimal convergence guarantees for the Riemannian corrected extragradient method (proposed in [1]) for geodesically strongly convex-strongly concave min-max optimization problems. A stochastic version of the algorithm is also discussed.

**Questions:**

My main question is on the applicability of the analysis as described in the previous section.

Also, I think the second part of Definition 2.2 since you have again convexity instead of concavity.

**Limitations:**

The authors discuss the limitations of their work. There is no direct societal impact.

**Strengths And Weaknesses:**

Strengths: The paper is very well-written in all its aspects and the technical work seems solid. The analysis answers a conjecture raised in [1], for an algorithm that is a reasonable adaptation of a similar Euclidean algorithm. Experimental results are also presented.

Weaknesses: Some parts of the proofs are hard to follow. Maybe this paper is a bit packed mathematically for the neurips community. My most serious concern is that the set of applications of geodesic strong convexity-strong concavity is exceptionally poor. Namely, the authors mention the problem of robust PCA as an application of their theoretical analysis, but this analysis cannot be applied to this problem because it is only *locally* strongly convex-strongly concave. This can have tremendous difference from *global* strong convexity-strong concavity, because one needs to ensure that an iterate of the algorithm remains in the set that the algorithm satisfies this property. This is usually impossible when the analysis is done for function values (Theorem 3.3) and for stochastic algorithms in general (Theorems 3.2,3.3). Even when the analysis is done for distances like in Theorem 3.1, one needs to ensure that $d_M(x_t,x^*) \leq d_M(x_0,x^*)$ and $d_N(y_t,y^*) \leq d_N(y_0,y^*)$ for all $t$. This is not clear to me because the contraction is only for the sum $d^2_M(x_t,x^*)+d^2_N(y_t,y^*)$. Thus, the paper is left without a concrete application of the theoretical analysis and this is the only reason I give a rejection score, which I would be happy to raise either if the authors can fix at least the analysis of their deterministic algorithm to cover also *local* strong convexity-strong concavity or can if they suggest a different important optimization problem that is actually *globally* strongly convex-strongly concave.

---

> ### Author Response · Authors · 2022-08-02
> **Reply to Reviewer LyY4**
>
> Thank you for your time and your input. Let us thank you also for the appreciation for the quality of the written draft. We hope that with our answer below we will convince you about the merits of our work. Below, we reply to your main questions point-by-point below, and we have colored all relevant revisions in our paper in $\color{blue}{\textrm{blue}}$:
>
> 1. **I would be happy to raise either if the authors can fix at least the analysis of their deterministic algorithm to cover also local strong convexity-strong concavity or can if they suggest a different important optimization problem that is actually globally strongly convex-strongly concave.**
> Thank you for your insightful comments! Let us first clarify why the current theoretical analysis of the deterministic algorithm covers local geodesic strong-convex-strong-concave settings. The key ingredient here is how to define the local region; indeed, if we say the set of $\{(x, y): d_M(x, x^\star) \leq \delta, d_N(y_t, y^\star) \leq \delta\}$ is a local region where the function is geodesic strong-convex-strong-concave. Then, the set of $\{(x, y): (d_M(x, x^\star)^2 + d_N(y_t, y^\star)^2) \leq \delta^2\}$ must be contained in the above local region and the objective function is geodesic strong-convex-strong-concave. If $(x_0, y_0) \in \{(x, y): (d_M(x, x^\star)^2 + d_N(y_t, y^\star)^2) \leq \delta^2\}$, our theoretical analysis guarantees the last-iterate linear convergencen rate. Such argument and definition of local region has become standard for min-max optimization in the Euclidean setting.
> Please see Assumption 2.1 in the reference: "Liang and Stokes, *Interaction matters: A note on non-asymptotic local convergence of generative adversarial networks*, AISTATS 2022".
>
>     For an important optimization problem that is actually globally geodesically strongly-convex-strongly-concave, we refer the reviewer to **Appendix B** where \textit{Robust matrix Karcher mean problem} is indeed the desired one. We apologize for the lack of examples in the main context and will put them back if more space is allowed for the final version.
>
> 2. **I think the second part of Definition 2.2 since you have again convexity instead of concavity.**
> We sincerely apologize for the typo and have corrected it in the revision.
>
> We thank you again for your detailed reading and your constructive input! We hope and trust that our replies have alleviated your concerns regarding the merits of our submission, and we look forward to an open-minded discussion if any such concerns remain.

---

> > ### Comment · Reviewer_LyY4 · 2022-08-09
> > **response which covers my concerns**
> >
> > Thank you for the good response, I revisited my scores accordingly. I feel that these points should be discussed more thoroughly in the main text and would invite the authors to do so if the paper gets accepted. For instance would be nice to know what $\delta$ should be in the case of rPCA, i.e. how close to the optima the algorithm needs to start.

---

### Official Review · Reviewer_rCdg · 2022-07-08

**Rating:** 7
**Confidence:** 3
**Soundness:** 2 fair
**Presentation:** 3 good
**Contribution:** 3 good

**Summary:**

The authors show that a previous algorithm from [1] (Sion’s theorem and algorithms, Zhang et al) achieves linear convergence rate for the last iterate in the (geodesically) strongly convex-strongly concave settings.

They also prove some guarantees under the assumption of a noisy gradient oracle, though of course a dependence on $1/\epsilon$ appears, as well as in the non-strongly convex case and non-smooth cases. These guarantees appear to match the best Euclidean guarantees apart from curvature parameters.

**Questions:**

1. Not sure if it's just me, but the figure in 3.1 is unreadable.
2. Can you give a reference for where RPCA is defined and used?
3. It might be appropriate to mention that in curvature -1, the condition number of g-convex functions has to depend on the radius and so sometimes the linear convergence rate is vacuous.
4. How were difficulties in previous upper bound proof attempts for the linear convergence guarantee overcome?

**Limitations:**

Yes.

**Strengths And Weaknesses:**

The strength of this paper is (1) the detailed description of the performance of Riemannian extragradient in various scenarios. The novel technical accomplishment is the proof of linear convergence rate in the strongly convex, strongly concave setting. The writing and clarity is high quality. It makes another recent result in [1], a  generalization of Sion's min-max theorem to the uniquely geodesic setting, more constructive.

I do take issue with the lack of examples mentioned for interesting Riemannian convex-convave problems. I think the reader has to take this on faith a bit. The main experimental example, robust PCA, needs Riemannian geometry to even formulate it. It would be nice to have a more natural example.

As one of the main strengths in the paper appears to be a proof which had heretofore eluded the community, it would help to mention what caused problems before and how the new proof overcomes those issues. Because I didn't find much discussion on this I have no way of knowing if the result is believable.

---

> ### Author Response · Authors · 2022-08-02
> **Reply to Reviewer rCdg**
>
> Thank you for your encouraging comments and positive evaluation! We reply to your main questions point-by-point below, and we have colored all relevant revisions in our paper in $\color{blue}{\textrm{blue}}$:
>
> 1. **Lack of examples in main draft.**
> We sincerely apologize for the lack of examples in the main context. Due to space limits, we have to defer many motivating examples to **Appendix B** where we provide a detailed justification. We will put them back if more space is allowed for the final version.
>
> 2. **Discussion on the proof sketch & Previous attempts.**
> Due to the lack of space, in our submission we decided to provide an intoduction about Riemannian min-max optimization for the broad NeurIPS audience and focus in the main text in a clear presentation of our results and the corresponding assumptions. In Lines 61-75, Page 2, however, we have explained clearly why it is challenging to derive the linear convergence rate of RCEG; More specifically, the previous proof in [1] attempts to directly bound $d_\mathcal{M}(x_{t+1}, x^\star)^2 + d_\mathcal{N}(y_{t+1}, y^\star)^2$. They claimed that this is intractable due to the nonlinear geometry of the manifold. Indeed, if somone would use the textbook analysis of ExtraGradient (See (Mokhtari et al., 2020) ) as a strategy pattern then the a direct comparsion between the term $d_\mathcal{M}(x_t, x^\star)^2 + d_\mathcal{N}(y_t, y^\star)^2$ and $d_\mathcal{M}(x_{t+1}, x^\star)^2 + d_\mathcal{N}(y_{t+1}, y^\star)^2$ should be done, which is indeed ''cursed'' with distortion phenomena due to the non-flat geometry (See Ahn and Sra (2020).).
>
>     In our analysis, we avoid this comparsion between these terms  but use them to bound a gap function defined by $f(\hat{x}_t, y^\star) - f(x^\star, \hat{y}_t)$ and some other terms. Since the objective function is geodesically strongly-convex-strongly-concave, we have $f(\hat{x}_t, y^\star) - f(x^\star, \hat{y}_t)$ is lower bounded by $\frac{\mu}{2}(d_\mathcal{M}(\hat{x}_t, x^\star)^2 + d_\mathcal{N}(\hat{y}_t, y^\star)^2)$. Then, using the relationship between $(x_t, y_t)$ and $(\hat{x}_t, \hat{y}_t)$, we conclude the desired results in Theorem 3.1. Notably, our approach is not affected by the nonlinear geometry of the manifold.
>
>     We would be happy to add in the revision draft an extensive explanation after the presentation of the results.
>
> 3. **Not sure if it's just me, but the figure in 3.1 is unreadable.**
> Interestingly, that was a distortion phenomenon due to the NeurIPS template.
> Our sincere apologies for this rescaling mistake which we have corrected in the revision version.
>
> 4. **Can you give a reference for where RPCA is defined and used?**
> Thank you for pointing it out. RPCA is a classical problem that is defined in the seminal work " *Robust principal component analysis?*" of Candes et. al. The problem in our paper is a geometry-aware extension of RPCA and we have cited the reference in the revision.
>
> 5. **It might be appropriate to mention that in curvature -1, the condition number of g-convex functions has to depend on the radius and so sometimes the linear convergence rate is vacuous.**
> Thank you for pointing out this subtle comment. If we understand correctly, you mean that $\tau_0$ will depend on the radius. This is the parameter that measures how non-flatness changes in the manifolds and intuitively will depend on the radius in certain cases. $\kappa$ is the condition number and our method has achieved the optimal dependence on it since our upper bound matches the lower bound in the Euclidean setting (where $\tau_0$ = 1). Nonetheless, we agree with you that sometimes the linear convergence rate is vacuous and it would be interesting to explore how to improve the dependence on $\tau_0$. For the interested reader, this is not a surprising issue. Similar phenomena recently has been discussed in the case of Riemannian Accelarated Gradient descent in geodesic convex minimization. While there is a range of parameter which RAGD is possible to provide better rates (See *Towards Riemannian Accelerated Gradient Methods*,Hongyi Zhang, Suvrit Sra), a recent work of Christopher Criscitiello, Nicolas Boumal (*Negative curvature obstructs acceleration for geodesically convex optimization, even with exact first-order oracles*
> ) showed that for specific choice of parameters Riemannian non-flat geometry can not disables any riemannian correction's power.
>
>     While our work established the positive side of the problem (falsifying the conjecture of [1]), a similar separation result for RCEG would be relatively interesting but also beyond the scope of the current paper, so we did not undertake it.

---

> > ### Comment · Reviewer_rCdg · 2022-08-06
> > **thank you for your comments.**
> >
> > Eom.

---

### Official Review · Reviewer_Zici · 2022-07-11

**Rating:** 6
**Confidence:** 3
**Soundness:** 3 good
**Presentation:** 3 good
**Contribution:** 3 good

**Summary:**

The paper analyzes first-order methods for min-max optimization on Riemannian manifolds. Problems of this form arise in a variety of settings, including optimal transport and adversarial learning. The main focus of the paper is on investigating the existence of a fundamental "performance gap" between Riemannian and Euclidean approaches for min-max optimization. The authors show that a Riemannian method (RCEG) matches the performance of its Euclidean counterpart in the geodesically strongly convex-concave case.

**Questions:**

See limitations.

**Limitations:**

1. Since the paper analyzes potential performance differences of Euclidean and Riemannian methods, it would have been good to compare against Euclidean methods in the experiments.
2. The theory is limited to the geodesically strongly convex-concave case. How restrictive is this? E.g., among the examples listed in the introductions, to which does the theory apply? Can you comment on any challenges in extending the present theory to less restrictive cases, which are understood in the Euclidean setting?
3. In sec. 4, l. 300, what does "appears to be locally geodesically strongly convex-concave" mean? Can this be shown?

**Strengths And Weaknesses:**

*Strength*
The paper addresses an important problem, which arises frequently in machine learning applications and is therefore very timely. The paper carefully reviews the existing literature and does a good job of explaining its contribution, especially with respect to Zhang et al. (2022), on which it directly builds.

*Weaknesses*
See limitations below.

---

> ### Author Response · Authors · 2022-08-02
> **Reply to Reviewer  Zici**
>
> Thank you for your encouraging comments and positive evaluation! We reply to your main questions point-by-point below, and we will color all relevant revisions in our paper in $\color{blue}{\textrm{blue}}$:
>
> 1. **Since the paper analyzes potential performance differences of Euclidean and Riemannian methods, it would have been good to compare against Euclidean methods in the experiments.**
> The Riemannian methods are in general the extension of Euclidean methods where the standard orthogonal projection onto manifolds is neither computationally efficient nor preserve nice properties. The RECG method is a generalization of the classical extragradient method, which is known to be unable to solve Riemannian min-max optimization. For the standard min-max optimization problems in Euclidean space, our method will reduce to extragradient methods whose empirical performance has been studied thoroughly; please see the intro and related work parts for the references.
> To make it even more precise, without the Exponential Map and the Riemannian Correction any Euclidean method is cursed to be out of any non-flat manifold after the first step.
>
> 2. **The theory is limited to the geodesically strongly convex-concave case. How restrictive is this? E.g., among the examples listed in the introductions, to which does the theory apply? Can you comment on any challenges in extending the present theory to less restrictive cases, which are understood in the Euclidean setting?**
> The last-iterate linear convergence rate in terms of Riemannian metrics is limited to geodesically strongly convex-concave cases but other results, e.g., the average-iterate sublinear convergence rate, are derived under more mild conditions. This is consistent with the classical results in the Euclidean setting where geodesic convexity reduces to convexity; indeed, the last-iterate linear convergence rate in terms of squared Euclidean norm is only known for strongly convex-concave cases. As such, our setting is not very restrictive.
>
>     Further, [1] shows that the existence of a global saddle point is only guaranteed under the geodesically convex-concave assumption. This is consistent with the classical results in the Euclidean setting. For geodesically nonconvex-concave or geodesically nonconvex-nonconcave cases, a global saddle point might not exist and new optimality notions are required before algorithmic design. This question remains open in the Euclidean setting and is beyond the scope of this paper. However, we remark that an interesting class of robustification problems are nonconvex-nonconcave min-max problems in the Euclidean setting can be geodesically convex-concave in the Riemannian setting. This exactly motivates our work. Such examples can be found in **Section B of Appendix**.
>
> 3. **What does "appears to be locally geodesically strongly convex-concave" mean? Can this be shown?**
> We apologize for our confusing words. This problem is indeed locally geodesically strongly convex-concave as argued in [1]. The proof can be found in (Zhang and Sra, 2016).

---

> > ### Comment · Reviewer_Zici · 2022-08-09
> > **Response to author reply**
> >
> > Thanks to the authors for their reply. After reading all reviews and your response I am keeping my score of acceptance.

---

### Official Review · Reviewer_KFH3 · 2022-07-11

**Rating:** 7
**Confidence:** 4
**Soundness:** 4 excellent
**Presentation:** 4 excellent
**Contribution:** 4 excellent

**Summary:**

This paper provides an extensive analysis of the Riemannian counterparts of Euclidean optimal first-order methods adapted to the manifold-constrained setting and  proves that the Riemannian corrected extragradient (RCEG) method achieves last-iterate convergence at a linear rate in the geodesically strongly- convex-concave case and extents it to the stochastic or non-smooth case.Some experiments have verified theoretical results.



**Questions:**

1. The first figure in Section 3 is not very clear and lacks labels;
2. In the experimental section, It would be good if there were more comparison algorithms.

**Ethics Review Area:**

["I don’t know"]

**Limitations:**

1. The first figure in Section 3 is not very clear and lacks labels;
2. In the experimental section, It would be good if there were more comparison algorithms.

**Strengths And Weaknesses:**

Overall, I think the paper is solid and novel enough. The authors prove that the Riemannian corrected extragradient (RCEG) method achieves last-iterate convergence at a linear rate in the geodesically strongly convex-concave case, matching the Euclidean result. Their results also extend to the stochastic or non-smooth case where RCEG and Riemanian gradient ascent descent (RGDA) achieve near-optimal convergence rates up to factors depending on curvature of the manifold.

---

> ### Author Response · Authors · 2022-08-02
> **Reply to Reviewer KFH3**
>
> Thank you for your encouraging comments and positive evaluation! We reply to your main questions point-by-point below, and we will color all relevant revisions in our paper in $\color{blue}{\textrm{blue}}$:
>
> 1. **The first figure in Section 3 is not very clear and lacks labels.**
> Interestingly, that was a distortion phenomenon due to the NeurIPS template. Our sincere apologies for this rescaling mistake which we have corrected in the revision version.
>
>
> 2. **In the experimental section, It would be good if there were more comparison algorithms.**
> Thank you for pointing out this. Riemannian min-max optimization is a new topic and RECG/RGDA are the only first-order method with theoretical guarantee, which are also parallel-transport free. We would appreciate if the reviewer could point out any references that we might have overlooked.

---

### Official Review · Reviewer_3XLa · 2022-07-13

**Rating:** 5
**Confidence:** 3
**Soundness:** 1 poor
**Presentation:** 1 poor
**Contribution:** 2 fair

**Summary:**

The authors explore the complexity of min-max problems defined on Riemannian manifolds. They prove, among others, linear convergence of the last iterate of RCEG, a result that was unknown in the literature, and explore stochastic and nonsmooth variants.

**Questions:**

- Which nonconvex constraints do appear in GANs? (first paragraph of the introduction)
- on p. 2, line 77-78, "for the non-convex non-concave setting, deriving ...for the convex case". What is convex here?
- please add the definition of the acronyms in the table in Section 1.1
- how do you define the diameter of a manifold, in Prop. 2.1? Are you restricting here youselves to a closed manifold?)
- Def. 2.2: shouldn't the second inequality be in the opposite direction?
- I am quite concerned about the unicity of geodesics assumption. In particular, this seems to rule out many manifolds of interest, including those of positive curvature (the sphere, the Stiefel manifold, the orthogonal group). Am I right to understand that this assumption is violated by your experimental section (see eq. 5)?
- How does this unicity assumption interact with your assumption on the sectional curvature, in regard of the previous comment?
- In Thm. 3.1, do kappa_min and kappa_max refer to the curvature of M or of N?


**Limitations:**

I do not expect this work to have any societal impact.

**Strengths And Weaknesses:**

The paper addresses an interesting topic. It is overall well-written, though there remains typos and inaccuracies at some places in the presentation. I am not very comfortable with the content and presentation of the results. Most of the paper is devoted to a presentation of the underlying assumptions of the convergence analysis, while those are already extensively presented in [1].  On the other hand, very little intuition is given to the new findings compared with [1] (understanding why the convergence guarantees for RCEG of this paper are stronger than in [1] would require to read in detail the appendix).
Given the fact that most of pages 1-2, 4-5 are already in [1], I would advise the authors to significantly shrink those sections, refering where applicable to [1], and give much more intuition on their main contributions:
- how do the authors achieve stronger guarantees than in [1] for RCEG?
- what are the changes needed to extend the proof to the sochastic setting?
- what are the changes needed to consider the nonsmooth setting?

---

> ### Author Response · Authors · 2022-08-02
> **Reply to Reviewer 3XLa**
>
> Thank you for your time and your input. We reply to your main questions point-by-point below, and we have colored all relevant revisions in our paper in $\color{blue}{\textrm{blue}}$:
>
> 1. **How do the authors achieve stronger guarantees than in [1] for RCEG?**
> Due to the lack of space, in our submission we decided to provide an intoduction about Riemannian min-max optimization for the broad NeurIPS audience and focus in the main text in a clear presentation of our results and the corresponding assumptions. In Lines 61-75, Page 2, however, we have explained clearly why it is challenging to derive the linear convergence rate of RCEG; More specifically, the previous proof in [1] attempts to directly bound $d_\mathcal{M}(x_{t+1}, x^\star)^2 + d_\mathcal{N}(y_{t+1}, y^\star)^2$. They claimed that this is intractable due to the nonlinear geometry of the manifold. Indeed, if somone would use the textbook analysis of ExtraGradient (See (Mokhtari et al., 2020) )as a strategy pattern then the a direct comparsion between the term $d_\mathcal{M}(x_t, x^\star)^2 + d_\mathcal{N}(y_t, y^\star)^2$ and $d_\mathcal{M}(x_{t+1}, x^\star)^2 + d_\mathcal{N}(y_{t+1}, y^\star)^2$ should be done, which is indeed ''cursed'' with distortion phenomena due to the non-flat geometry ( See Ahn and Sra (2020).  ).  \\\\
>
>     In our analysis, we avoid this comparsion between these terms  but use them to bound a gap function defined by $f(\hat{x}_t, y^\star) - f(x^\star, \hat{y}_t)$ and some other terms. Since the objective function is geodesically strongly-convex-strongly-concave, we have $f(\hat{x}_t, y^\star) - f(x^\star, \hat{y}_t)$ is lower bounded by $\frac{\mu}{2}(d_\mathcal{M}(\hat{x}_t, x^\star)^2 + d_\mathcal{N}(\hat{y}_t, y^\star)^2)$. Then, using the relationship between $(x_t, y_t)$ and $(\hat{x}_t, \hat{y}_t)$, we conclude the desired results in Theorem 3.1. Notably, our approach is not affected by the nonlinear geometry of the manifold.
>
> 2. **What are the changes needed to extend the proof to the stochastic setting?**
> For the stochastic setting, the key ingredient to get the optimal convergence rate is to carefully select the step size such that the noise of the gradient estimator will not affect the final convergence rate significantly. As a highlight, such technique has been used for analyzing stochastic RCEG in the Euclidean setting; see [86]. Our analysis can be seen as their extension to the Riemannian setting.
>
> 3. **What are the changes needed to consider the nonsmooth setting?**
> For the nonsmooth setting, we need to consider a different algorithm, namely RGDA, which is a direct extension of classical gradient descent ascent in the Euclidean setting. Indeed, GDA was known as optimal in a nonsmooth Euclidean setting and it is natural for us to study its generalization. The analysis is relatively simpler compared to smooth settings but we still need to deal with the issue caused by the nonlinear geometry of manifolds and the interplay between the distortion of Riemannian metrics, the gap function and the bounds of Lipschitzness of our bi-objective.
>
> $\color{blue}{\textrm{To make all this clear, we have included this discussion in the revised version of our paper.}}$
>
> 4. **Which non-convex constraints do appear in GANs? (first paragraph of the introduction)?**
> This is a good catch. It depends of the way of formulating the problem. Of-course if the optimization problem is over the parameters of the neural-nets, the problem is non-convex non-concave but without complex constraints. On the other hand, if the formulation of the problem is over the distribution of images over some population and the set of all different discriminator, manifolds appear naturally (See Prof. Alex Dimakis' talk in  [AI Texas Summit 2022, Slide 21](https://users.ece.utexas.edu/~dimakis/Dimakis_TexasAI.pdf) )
> Nonetheless, we would of course be happy to remove it to avoid legitimate misunderstanding in a further revision if the program committee requested it.
>
> 5. **Page 2, line 77-78, "for the non-convex non-concave setting, deriving ...for the convex case". What is convex here?**
> Let us rephrase more extensively the aforementioned sentence: Huang et al. [79] analyzed the Riemannian GDA (RGDA) for the non-convex non-concave setting. Therefore, in their analysis they can not obtain actually last-iterate convergence results and even in the average/best iterate setting due to the lack of the machinery that convex analysis & optimization offers they derive sub-optimal rates for the geodesic convex-concave case, which is the problem of our interest.
>
> 6.  **Please add the definition of the acronyms in the table in Section 1.1.**
> Thank you for pointing it out and we have added the definition of acronyms in the revision.

---

> ### Author Response · Authors · 2022-08-02
> **Reply to Reviewer 3XLa (Continued)**
>
> 7.  **How do you define the diameter of a manifold, in Prop. 2.1? Are you restricting here to a closed manifold?)**
> In Prop 2.1, the diameter is defined as the supremum of the distance between two points at the manifold under Riemannian metric. Our manifold may not be closed; indeed, if the upper bound on the sectional curvature is positive, we require the manifold to be compact such that the main inequality holds; see Corollary 2.1 in Alimisis et al. (2020). However, if the upper bound is non-positive, the diameter of the manifold can be infinite but the main inequality still holds with $\underline{\xi} = 1$. As such, our analysis covers the Hadamard manifold.
>
>
> 8.  **Def. 2.2: shouldn't the second inequality be in the opposite direction?**
> We sincerely apologize for the typo and have corrected it in the revision.
>
> 8.  **I am quite concerned about the unicity of geodesics assumption. In particular, this seems to rule out many manifolds of interest, including those of positive curvature (the sphere, the Stiefel manifold, the orthogonal group). Am I right to understand that this assumption is violated by your experimental section (see Eq. 5)?**
> The unicity of geodesics assumption is algorithm-independent and is imposed here for guaranteeing that the geodesic convex concave Riemannian problems always admit saddle-point solutions [1]. Even though this rules out many manifolds of interest as R1 points out, there are still many manifolds that satisfy such conditions. More specifically, the Hadamard manifold (manifolds with non-positive curvature, $\kappa_\max = 0$) admits a unique geodesic between any two points. As such, this is a common regularity condition in Riemannian optimization and has become standard in the literature; see (Zhang and Sra, 2016) and (Alimisis et al., 2020).
>
>     We apologize for our inaccurate wording in the experimental section. Indeed, our problem satisfies \textbf{most of} the assumptions we impose in this paper: the SPD manifold is Hadamard but the sphere manifold is a complete but \textbf{not unique} manifold. The reasons why we use such example are
>     + (i) it is a classical one in ML;
>     + (ii) [1] also uses this example and observes the linear convergence behavior;
>     + (iii) the numerical results show that the unicity of geodesics assumption may not be necessary in practice;
>     + (iv) this is an application where both min and max sides are done on Riemannian manifolds.
>
>     Nonetheless, we agree with you and will try some examples that meet the conditions in the final version, e.g., Distributionally robust Riemannian optimization or Robust matrix Karcher mean problem (Appendix B).
>
> 10. **How does this unicity assumption interact with your assumption on the sectional curvature, in regard to the previous comment?**
> Thank you for pointing out this. In Assumption 2.1, the third point in fact implies the second point. Indeed, an upper-bound on sectional curvature guarantees that the manifold is geodesically complete and unique. We highlight the unicity assumption to make our content more accessible to the reader who does not have much background on Riemannian optimization.
>
> 11. **In Thm. 3.1, do $\kappa_{min}$ and $\kappa_{max}$ refer to the curvature of M or of N?**
> Yes. Thm. 3.1 is derived under Assumption 2.1 where $\kappa_{min}$ and $\kappa_{max}$ are assumed to be the lower bound and upper bound of section curvatures of both $\mathcal{M}$ and $\mathcal{N}$.
>
> Thanks again for your remarks! We hope and trust that our replies have alleviated your concerns regarding the merits of our submission, and we look forward to an open-minded discussion if any such concerns remain.

---

### Meta-Review · Area_Chair_hmpM · 2022-08-24

**Recommendation:** Accept
**Confidence:** Certain

**Metareview:**

The paper analyzes the performance of a few of the recent algorithms for min-max optimization over manifolds. The analysis is extensive and is interesting in its own respect. In the final version, please address the comments of the reviewers.



**Award:**

No

---

### Decision · Program_Chairs · 2022-09-14

Accept